# ResCP: Reservoir Conformal Prediction for Time Series Forecasting

**Roberto Neglia**[*]
UiT the Arctic University of Norway

**Andrea Cini**[*]
IDSIA USI-SUPSI, Università della Svizzera italiana
Swiss National Science Foundation Postdoctoral Fellow

**Michael M. Bronstein**
University of Oxford
AITHYRA

**Filippo Maria Bianchi**
UiT the Arctic University of Norway
NORCE Norwegian Research Centre AS

## ABSTRACT

Conformal prediction (CP) offers a powerful framework to build distribution-free prediction intervals for exchangeable data. Existing methods that extend CP to sequential data rely on fitting a relatively complex model to capture temporal dependencies. However, these methods can fail if the sample size is small and often require expensive retraining when the underlying data distribution changes. To overcome these limitations, we propose *Reservoir Conformal Prediction* (ResCP), a novel training-free CP for time series. Our approach leverages the efficiency and representation capabilities of Reservoir Computing to dynamically reweight conformity scores. In particular, we compute similarity scores across reservoir states and use them to adaptively reweight the observed residuals. ResCP enables us to account for local temporal dynamics when modeling the error distribution without compromising computational scalability. We prove that, under reasonable assumptions, ResCP achieves asymptotic conditional coverage, and we empirically demonstrate its effectiveness across diverse forecasting tasks.

## 1 INTRODUCTION

Despite deep learning often achieving state-of-the-art results in time series forecasting, widely used methods do not provide a way to quantify the uncertainty of the predictions (Benidis et al., 2022), which is crucial for their adoption in risk-sensitive scenarios, such as healthcare (Makridakis et al., 2019), load forecasting (Hong & Fan, 2016), and weather forecasting (Palmer & Hagedorn, 2006). Moreover, in many cases, prediction intervals (PIs) must not only be reliable but also fast to compute. Existing uncertainty quantification approaches for time series often rely on strong distributional assumptions (Benidis et al., 2022; Salinas et al., 2020), which do not always fit real-world data. Moreover, they often require long and expensive training procedures, as well as modifications to the underlying forecasting model, which limits their applicability to large datasets. A framework that recently gained attention in uncertainty quantification is CP (Vovk et al., 2005; Angelopoulos et al., 2023). CP assumes exchangeability between the data used to build the PIs and the test data points, meaning that the joint distribution of the associated sequence of random variables does not change when indices are permuted. Clearly, this does not hold for time series, as the presence of temporal dependencies and, possibly, non-stationarity violates the assumption. Moreover, temporal dynamics can be heterogeneous and often result in heteroskedastic errors. This requires mechanisms to make PIs locally adaptive (Lei et al., 2018; Guan, 2022).

To extend CP to time series data, a popular approach is to reweight the observed residuals to handle non-exchangeability and/or temporal dependencies (Barber et al., 2023; Tibshirani et al., 2019; Auer et al., 2023). In particular, Auer et al. (2023) proposes an effective reweighting mechanism based on a learned model and soft-attention operators, which, however, suffers from the high computational cost typical of Transformer-like architectures. Other methods build PIs by directly learning the

---

[*]Equal contribution. Correspondance to: `roberto.neglia@uit.no`; `andrea.cini@usi.ch`.

quantile function of the error distribution (Jensen et al., 2022; Xu & Xie, 2023a; Lee et al., 2024; 2025; Cini et al., 2025). This, however, often results in high sample complexity and might require frequent model updates to adapt to changes in the data distribution.

To address these limitations, we propose *Reservoir Conformal Prediction* (RESCP), a novel CP method that leverages a reservoir, i.e., a randomized recurrent neural network (Jaeger, 2001; Lukoševičius & Jaeger, 2009), driven by the observed prediction residuals to efficiently compute data-dependent weights adaptively at each time step. To capture locally similar dynamics, the weights are derived from the similarity between current and past states of the reservoir, which is implemented as an echo state network (ESN) (Jaeger, 2001; Gallicchio & Scardapane, 2020). This enables RESCP to model local errors through a weighted empirical distribution of past residuals, where residuals associated with similar temporal dynamics receive higher weights. The ESN does not require any training and is initialized to yield stable dynamics and informative representations of the input time series at each time step. This makes our approach extremely scalable and easy to implement. RESCP can be applied on top of *any* point forecasting model, since it only requires residuals from a disjoint calibration set.

Our main contributions are summarized as follows.

- We provide the first assessment of reservoir computing for CP in time series analysis and show that it is a valid and scalable alternative to existing methods.

- We introduce RESCP, a novel, scalable, and theoretically grounded tool for distribution-free uncertainty quantification in sequential prediction tasks.

- We prove that RESCP can achieve asymptotic conditional coverage under reasonable assumptions on the data-generating process.

- We introduce variants of RESCP to handle distribution shifts and leverage exogenous variables.

We evaluate RESCP on data from real-world applications and show its robustness and effectiveness by comparing it to state-of-the-art baselines.

## 2 BACKGROUND AND RELATED WORK

We consider a forecasting setting where the goal is to predict future values of a time series based on past observations. Let $\{y_t\}_{t=1}^T$ be the sequence of scalar target observations at each time step $t$. We denote by $y_{1:T}$ the entire history of observations up to time $T$. Moreover, let $\{\boldsymbol{u}_t\}_{t=1}^T$ be the sequence of exogenous variables at each time step $t$ of dimension $D_u$. In this context, we are also given a forecasting model $\hat{y}_{t+H} = \hat{f}(y_{t-W:t}, \boldsymbol{u}_{t-W:t})$ that produces point forecasts of $H$-steps-ahead, with $H \geq 1$, given a window $W \geq 1$ of past observations. The model $\hat{f}$ can be any kind of forecasting model, e.g., a recurrent neural network (RNN). The objective is to construct valid PIs that reflect the uncertainty in the forecasts. Ideally, PIs should be conditioned on the current state of the system, i.e., we want to achieve conditional coverage:

$$\mathbb{P}\left(y_{t+H} \in \hat{C}_T^\alpha(\hat{y}_{t+H}) \big| y_{\leq t}, \boldsymbol{u}_{\leq t}\right) \geq 1 - \alpha \tag{1}$$

where $\hat{C}_T^\alpha(\hat{y}_{t+H})$ is the prediction interval for the forecast $\hat{y}_{t+H}$ at significance level $\alpha$.

### 2.1 CONFORMAL PREDICTION IN TIME SERIES FORECASTING

CP (Vovk et al., 2005; Angelopoulos et al., 2023) is a framework for building valid *distribution-free* PIs, possibly with finite samples. CP provides a way to quantify uncertainty in predictions by leveraging the empirical quantiles of *conformity* scores, which in our regression setting we define as the prediction residuals

$$r_t = y_t - \hat{y}_t. \tag{2}$$

These scores measure the discrepancy between the model predictions and the observed values. The CP framework we consider in this work is split CP (SCP) (Vovk et al., 2005), which constructs valid PIs post-hoc by leveraging a disjoint calibration set to estimate the distribution of prediction errors. In its standard formulation, CP treats all calibration points symmetrically when building PIs. In heterogeneous settings, this can lead to overly conservative intervals as conformal scores

are often heteroskedastic. To address heteroskedasticity, CP can be made *locally adaptive* by providing PIs whose width can adaptively shrink and inflate at different regions of the input space (Lei et al., 2018). In this context, Guan (2022) proposed localized CP (LCP), which reweights conformity scores according to the similarity between samples in feature space, given a localization function. Hore & Barber (2023) introduced a randomized version of LCP, with approximate conditional coverage guarantees and more robust under covariate shifts. Although this reweighting breaks exchangeability, validity is recovered through a finite-sample correction of the target coverage level. Other approaches rely on calibrating estimates of the spread of the data (Lei & Wasserman, 2013) or quantile regression (Romano et al., 2019; Jensen et al., 2022; Feldman et al., 2023), but often require ad-hoc base predictors (i.e., they cannot be applied on top of generic point predictors).

**Conformal prediction for time series**  Under the assumption of data exchangeability, CP methods provide, in finite samples, valid intervals given a specified level of confidence (Angelopoulos et al., 2023). However, in settings where exchangeability does not hold, standard CP methods can fail in providing valid coverage (Barber et al., 2023; Tibshirani et al., 2019). This is the case of time series data where temporal dependencies break the exchangeability assumption. Barber et al. (2023) show that reweighting the available residuals (in a data-independent fashion) can handle non-exchangeability and distribution shifts, and introduce NexCP, a weighting scheme that exponentially decays the past residuals over time. Barber & Tibshirani (2025) provides a unified view of several weighted CP methods as approaches to condition uncertainty estimates on the available information on the target data point. Other recent approaches, instead, fit a model to account for temporal dependencies and rely on asymptotic guarantees under some assumptions on the data-generating process (Xu & Xie, 2023a; Auer et al., 2023; Lee et al., 2025). For example, SPCI (Xu & Xie, 2023a) uses a quantile random forest trained at each step on the most recent scores. Conversely, HopCPT leverages a Hopfield network to learn a data-dependent reweighting scheme based on soft attention (Auer et al., 2023). Lee et al. (2025) use a reweighted Nadaraya-Watson estimator to perform quantile regression over the past nonconformity scores and derive the weights from the kernel function. Cini et al. (2025) proposes instead to use graph neural networks (Bacciu et al., 2020; Bronstein et al., 2021) to additionally condition uncertainty estimates on correlated time series. As already mentioned, we focus on methods that estimate uncertainty on the prediction of a pre-trained model; we refer to Benidis et al. (2022) for a discussion of probabilistic forecasting architectures.

## 2.2 ECHO STATE NETWORKS

ESNs are among the most popular reservoir computing (RC) methods (Jaeger, 2001; Lukoševičius & Jaeger, 2009). In particular, ESNs are RNNs that are randomly initialized to ensure the stability and meaningfulness of the representations and left untrained. An ESN encodes input sequences into nonlinear, high-dimensional state representations through the recurrent component, called *reservoir*, which extrapolates from the input a rich pool of dynamical features. The state update equation of the reservoir is

$$\boldsymbol{h}_t = (1 - l)\boldsymbol{h}_{t-1} + l\,\sigma\left(\boldsymbol{W}_x\boldsymbol{x}_t + \boldsymbol{W}_h\boldsymbol{h}_{t-1} + \boldsymbol{b}\right), \tag{3}$$

where $\boldsymbol{x}_t$ denotes the input at time $t$, $\boldsymbol{W}_x \in \mathbb{R}^{D_h \times D_x}$ and $\boldsymbol{W}_h \in \mathbb{R}^{D_h \times D_h}$ are fixed random weight matrices, $\boldsymbol{b} \in \mathbb{R}^{D_h}$ is a random bias vector, $\boldsymbol{h}_t \in \mathbb{R}^{D_h}$ is the reservoir state, $l \in (0, 1]$ is the leak rate which controls how much of the current state is retained at each update, and $\sigma$ is a nonlinear activation, typically the hyperbolic tangent. When $\boldsymbol{W}_x$ and $\boldsymbol{W}_h$ are properly initialized (see,e.g., Lukoševičius 2012), reservoir states $\boldsymbol{h}_t$ provide expressive embeddings of the past inputs $\boldsymbol{x}_{\leq t}$, which can be processed by a trainable readout to perform downstream tasks such as time series classification or forecasting (Bianchi et al., 2020). Since embeddings already model nonlinear dynamics, the readout is usually implemented as a simple linear layer which can be trained efficiently.

**Echo state property**  The reservoir dynamics should be neither too contractive nor chaotic. In the first case, the reservoir produces representations that are not sufficiently expressive. In the latter, the reservoir is unstable and responds inconsistently to nearly identical input sequences. ESNs are usually configured to ensure the Echo State Property (ESP), a necessary condition for global asymptotic stability under which the reservoir state asymptotically forgets its initial conditions and the past inputs. Formally, for any initial states $\boldsymbol{h}_0, \boldsymbol{h}_0'$ and input sequence $\boldsymbol{x}_{1:T}$, by calling $f_R(\boldsymbol{h}_0, \boldsymbol{x}_{1:T})$ the final state of the reservoir initialized with $\boldsymbol{h}_0$ and being fed with $\boldsymbol{x}_{1:T}$, then the ESP is defined as

$$||f_R(\boldsymbol{h}_0, \boldsymbol{x}_{1:T}) - f_R(\boldsymbol{h}_0', \boldsymbol{x}_{1:T})|| \to 0 \ \text{ as } \ T \to \infty \tag{4}$$

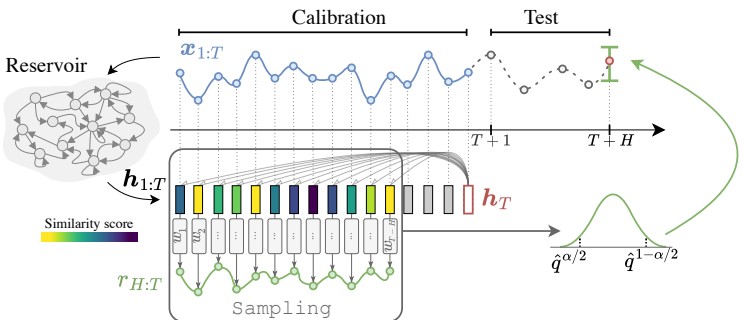

Figure 1: Let $\{\boldsymbol{x}_t\}_{t=1}^T$ and $\{r_t\}_{t=1}^T$ be the time series and residuals of the calibration set, respectively. Each sequence $\boldsymbol{x}_{1:t}$ generates the state of the reservoir $\boldsymbol{h}_t$. The last state $\boldsymbol{h}_T$ is the query state for the prediction of $\hat{y}_{T+H}$. We compute similarity scores between $\boldsymbol{h}_T$ and the calibration states $\{\boldsymbol{h}_t\}_{t=1}^{T-H}$, which are used to reweight with $\{w_t\}_{t=1}^{T-H}$ associated residuals $\{r_t\}_{t=H}^T$ by resampling them. Quantiles are computed from the sampled residuals and used to build the PI for $\hat{y}_{T+H}$.

The ESP can be achieved by properly initializing $\boldsymbol{W}_h$ and by setting its spectral radius $\rho(\boldsymbol{W}_h) < 1$ (Yildiz et al., 2012; Gallicchio & Scardapane, 2020).

## 3 RESERVOIR CONFORMAL PREDICTION

We propose a novel approach for computing CP intervals for time series forecasting based on the representations generated by RC to model temporal dependencies. Our method leverages CP to provide accurate uncertainty estimates while maintaining the computational efficiency of RC.

### 3.1 RESERVOIR SAMPLING CONFORMAL PREDICTION

We construct PIs from a calibration set consisting of a time series of residuals of length $T$. In particular, we aim at reweighting the residuals based on similarities with the local dynamics at the target time step. We leverage the reservoir capability of embedding the observed dynamics $\boldsymbol{x}_{\leq t}$ into its state $\boldsymbol{h}_t$. An overview of the whole procedure is depicted in Figure 1. In general, the input $\boldsymbol{x}_t$ can include any set of endogenous and exogenous variables available at time step $t$. In the following, unless differently stated, we consider $\boldsymbol{x}_t$ to be the prediction residual ($\boldsymbol{x}_t = r_t$ obtained by training a generic forecasting model on a disjoint training set); the discussion on how to effectively incorporate exogenous variables is deferred to Section 3.2.

Given an ESN with dynamics defined in Equation 3 and hyperparameters $\boldsymbol{\theta}$, the time series $\boldsymbol{x}_{1:t} = (\boldsymbol{x}_1, \ldots, \boldsymbol{x}_t)$ is embedded into a sequence of $t$ reservoir states which encode observed dynamics at each time step as

$$\boldsymbol{h}_t = \text{ESN}_{\boldsymbol{\theta}}(\boldsymbol{x}_{1:t}), \qquad \boldsymbol{h}_t \in \mathbb{R}^{D_h}. \tag{5}$$

In particular, to compute the PI for the $H$-step-ahead prediction $\hat{y}_{t+H}$, we take as *query state* the current state $\boldsymbol{h}_t$ (since we do not have access to states $\boldsymbol{h}_{>t}$) and compute similarity scores between $\boldsymbol{h}_t$ and the states in the calibration set. Note that in most scenarios, the calibration set can be iteratively updated as new observations become available (i.e., we can include in the calibration set all the observations up to the current time step $t$). Weights are computed as

$$z_s(\boldsymbol{h}_t) = \text{SIM}(\boldsymbol{h}_t, \boldsymbol{h}_s), \qquad 1 \leq s \leq T - H \tag{6}$$

$$\{w_1(\boldsymbol{h}_t), \ldots, w_{T-H}(\boldsymbol{h}_t)\} = \text{SOFTMAX}\left\{\frac{z_1(\boldsymbol{h}_t)}{\tau}, \ldots, \frac{z_{T-H}(\boldsymbol{h}_t)}{\tau}\right\} \tag{7}$$

where SIM is a similarity score (e.g., dot product) and $\tau > 0$ is a temperature hyperparameter. Our goal is to approximate the conditional cumulative distribution function (CDF) of the residual at time $t + H$ with

$$\widehat{F}(r \mid \boldsymbol{h}_t) := \sum_{s=1}^{T-H} w_s(\boldsymbol{h}_t) \mathbb{1}\big(r_{s+H} \leq r\big) \approx \mathbb{P}(r_{t+H} \leq r \mid \boldsymbol{x}_{\leq t}) \tag{8}$$

where $w_s(\boldsymbol{h}_t)$ is a weight proportional to the similarity of $\boldsymbol{h}_s$ to $\boldsymbol{h}_t$ and such that $\sum_s w_s(\boldsymbol{h}_t) = 1$. Quantiles of the conditional error distribution can then be estimated as:

$$\widehat{Q}_\beta(\boldsymbol{h}_t) := \inf\{r \in \mathbb{R} \ : \ \widehat{F}(r \mid \boldsymbol{h}_t) \geq \beta\}. \tag{9}$$

In practice, instead of computing the weighted quantile in Eq. 9 directly, we approximate it through Monte Carlo sampling akin to Auer et al. (2023). More specifically, we sample residuals accordingly to the weights in Eq. 7 and compute the standard empirical $\beta$-quantile on the sampled residuals:

$$\hat{q}_{t+H}^{\alpha/2} = \widehat{Q}_{\alpha/2}(\boldsymbol{h}_t), \qquad \hat{q}_{t+H}^{1-\alpha/2} = \widehat{Q}_{1-\alpha/2}(\boldsymbol{h}_t). \tag{10}$$

PIs with the desired confidence level $\alpha$ are obtained as:

$$\widehat{C}_T^\alpha(\hat{y}_{t+H}) = \left[\hat{y}_{t+H} + \hat{q}_{t+H}^{\alpha/2}, \hat{y}_{t+H} + \hat{q}_{t+H}^{1-\alpha/2}\right]. \tag{11}$$

To account for skewed error distributions, we follow the approach of Xu & Xie (2023b) and refine the interval in 11 by selecting the level $\beta^*$ that minimizes its width:

$$\beta^* = \underset{\beta \in [0,\alpha]}{\arg\min} \left[\widehat{Q}_{1-\alpha+\beta}(\boldsymbol{h}_t) - \widehat{Q}_\beta(\boldsymbol{h}_t)\right]$$

$$\widehat{C}_T^\alpha(\hat{y}_{T+H}) = \left[\hat{y}_{T+H} + \widehat{Q}_{\beta^*}(\boldsymbol{h}_t), \hat{y}_{T+H} + \widehat{Q}_{1-\alpha+\beta^*}(\boldsymbol{h}_t)\right]. \tag{12}$$

This can provide narrower PIs, at the cost of searching for the optimal level $\beta$ (Xu & Xie, 2023b).

**Discussion** Essentially, RESCP is a local CP method for time series where local similarity is gauged by relying on reservoir states. By sampling more residuals associated with similar states, we condition the estimates on specific dynamics of interest in the time series. This allows us to build locally adaptive intervals. While Guan (2022) provides asymptotic guarantees for standard local LCP for i.i.d. data, such results cannot be applied in the context of time series. In the following section, we discuss the conditions that allow RESCP to provide (asymptotically) valid intervals. Notably, this will require assumptions on the nature of the process generating the data and on the reservoir dynamics. Finally, note that with finite samples, even with i.i.d. observation, data-dependent weights might introduce a bias that should be accounted for (Guan, 2022). In practice, we found that tuning RESCP hyperparameters on a validation set was sufficient to provide accurate coverage in most scenarios (see Sec. 4).

### 3.1.1 THEORETICAL ANALYSIS

We analyze the theoretical properties of RESCP, and show in Corollary 3.7 that it asymptotically guarantees coverage under some regularity assumptions. As already mentioned, guarantees of standard CP methods break down for time series unless one relies on highly unrealistic assumptions (e.g., exchangeable or i.i.d. observations) (Barber et al., 2023). Moreover, even assuming exchangeability, it is impossible to construct finite-length PIs with distribution-free conditional coverage guarantees (Lei & Wasserman, 2013; Vovk, 2012). We start by introducing the assumptions needed to prove the consistency of the weighted empirical CDF in Eq. 8.

**Assumption 3.1** (Time-invariant and mixing process)**.** *We assume that the process* $\{Z_t = (\boldsymbol{x}_t, r_{t+H})\}_{t=1}^\infty$ *is time-invariant and strongly mixing ($\alpha$-mixing) with coefficient $\alpha(k) \to 0$ as $t \to \infty$.*

Intuitively, Assumption 3.1 implies that the system's dynamics do not change over time and that it "forgets" its initial conditions and past structure, thereby allowing observations that are far apart in time to be treated as if they were independent.

**Assumption 3.2** (ESP and Lipschitz properties in ESN)**.** *Let $\boldsymbol{x}_{\leq t} = (\ldots, \boldsymbol{x}_{t-1}, \boldsymbol{x}_t)$ denote the history. Define the fading-memory metric for $\gamma \in (0,1)$ by*

$$d_{\mathrm{fm}}(\boldsymbol{x}_{\leq t}, \boldsymbol{x}'_{\leq t}) := \sum_{k=1}^\infty \gamma^k \|\boldsymbol{x}_{t-k+1} - \boldsymbol{x}'_{t-k+1}\|_2.$$

*Then the ESN state map $ESN_{\boldsymbol{\theta}} : \boldsymbol{x}_{\leq t} \mapsto \boldsymbol{h}_t \in \mathbb{R}^{D_h}$ is well-defined and causal with fading memory (i.e., the output only depends on the past and present inputs), and there exist a constant $L_X > 0$ such that for all histories $\boldsymbol{x}_{\leq t}, \boldsymbol{x}'_{\leq t}$:*

$$\|ESN_{\boldsymbol{\theta}}(\boldsymbol{x}_{\leq t}) - ESN_{\boldsymbol{\theta}}(\boldsymbol{x}'_{\leq t})\|_2 \ \leq \ L_X \, d_{\mathrm{fm}}(\boldsymbol{x}_{\leq t}, \boldsymbol{x}'_{\leq t}).$$

Assumption 3.2 tells us that reservoir has stable and contractive dynamics: past inputs influence the state in a controlled, exponentially decaying way, and small perturbations are eventually washed out from the reservoir's state. These assumptions are commonly used in RC (Grigoryeva & Ortega, 2019; Gallicchio & Micheli, 2011).

**Assumption 3.3** (Continuity of the conditional law). *Let $\mathbb{P}(r_{t+H} \in \mathbb{R} \mid \boldsymbol{x}_{\leq t})$ denote the conditional law of $r_{t+H}$ given the past. Then for every $r \in \mathbb{R}$ the map*

$$\boldsymbol{x}_{\leq t} \;\mapsto\; F(r \mid \boldsymbol{x}_{\leq t})$$

*is continuous in $\boldsymbol{x}_{\leq t}$ with respect to $d_{\mathrm{fm}}$, i.e.*

$$d_{\mathrm{fm}}(\boldsymbol{x}_{\leq t}, \boldsymbol{x}'_{\leq t}) \to 0 \implies \sup_r |F(r \mid \boldsymbol{x}_{\leq t}) - F(r \mid \boldsymbol{x}'_{\leq t})| \to 0$$

Intuitively, Assumption 3.3 ensures that if two histories are similar (in the fading-memory sense), then the distributions of the future residuals conditioned on these histories are also close. This is a reasonable assumption to enable learning.

**Definition 3.4** (Effective sample size). *Let $n$ be the number of available calibration samples, the effective sample size (Kong et al., 1994; Liu, 2008) is defined in our setting as*

$$m_n := \left( \sum_{i=1}^{n} (w_i^{(n)})^2 \right)^{-1}. \tag{13}$$

Note that at high temperature, weights are approximately uniform ($w_i^{(n)} \approx \frac{1}{n}$) and $\sum_i (w_i^{(n)})^2 \approx \frac{1}{n}$, hence $m_n \approx n$, which means that the method collapses to vanilla SCP as all calibration points are treated symmetrically. If instead the temperature is too low, then all the mass concentrates on only one point, meaning that $\sum_i (w_i^{(n)})^2 = 1$ and hence $m_n = 1$.

**Assumption 3.5** (Softmax weighting scheme). *Let $n$ be the number of available calibration samples, and let $\tau_n = \tau(n)$ be the temperature parameter of the SOFTMAX of equation 7, $m_n$ the associated effective sample size, and $\boldsymbol{w}^{(n)}$ the output weights. Assume:*

*(i) $\tau_n$ is configured to slowly decrease as $n \to \infty$, so that $m_n \to \infty$;*

*(ii) for every $\delta > 0$,*

$$\sum_{s:\|\boldsymbol{h}_s - \boldsymbol{h}_t\|_2 \geq \delta} w_s^{(n)}(\boldsymbol{h}_t) \;\xrightarrow[n\to\infty]{\mathbb{P}}\; 0.$$

Condition (ii) states that for sufficiently small temperature $\tau_n$, the softmax normalization should concentrate mass on those calibration points whose states lie in a shrinking neighborhood of $\boldsymbol{h}_t$. Conversely, we also need these points within the shrinking neighborhood of $\boldsymbol{h}_t$ to increase as the size of the calibration set increases. This can be seen as a *bias-variance tradeoff*. In practical terms, the temperature $\tau_n$ must be set to a small enough value to localize weights at points similar to the query state (reducing *bias*), but at the same time it should be large enough to guarantee a good effective sample size. This requirement translates into Assumption 3.5 (i) which prescribes that the temperature has to shrink at a reasonably slow rate: one that allows the number of effective neighbors $m_n$ to diverge as $n$ grows.

**Theorem 3.6** (Consistency of the weighted empirical CDF). *Let $\widehat{F}_n(\,\cdot\,\mid\boldsymbol{h}_t)$ denote the conditional weighted empirical CDF in Eq. 8 with calibration data $\{(\boldsymbol{x}_i, r_i)\}_{i=1}^n$ and $F_n(r \mid \boldsymbol{h}_t) := \mathbb{P}(r_{t+H} \leq r \mid \boldsymbol{h}_t)$. Under Assumptions 3.1–3.5, we have for any query state $\boldsymbol{h}_t \in \mathbb{R}^{D_h}$,*

$$\sup_{r\in\mathbb{R}} \left| \widehat{F}_n(r \mid \boldsymbol{h}_t) - F(r \mid \boldsymbol{h}_t) \right| \;\xrightarrow[n\to\infty]{\mathbb{P}}\; 0.$$

The proof can be found in App. A. Theorem 3.6 implies the consistency of the empirical conditional quantile estimator $\widehat{Q}_\beta(\boldsymbol{h}_t)$, defined in Equation 9, with respect to the true conditional quantile $Q_\beta(\boldsymbol{h}_t)$. Finally, it is trivial to show that the asymptotic coverage of RESCP is guaranteed.

**Corollary 3.7** (Asymptotic conditional coverage guarantee). *Under Assumptions 3.1–3.5, for any $\alpha \in (0, 1)$ and for $n \to \infty$,*

$$\mathbb{P}\left( y_{t+h} \in \hat{C}_t^\alpha(\hat{y}_{t+h}) \mid \boldsymbol{h}_t \right) \;\xrightarrow[n\to\infty]{\mathbb{P}}\; (1 - \alpha).$$

**Remark** In the theoretical analysis, we model the distribution of future residuals conditioned on the state $\boldsymbol{h}_t$, i.e., on the reservoir state at time $t$. This is clearly weaker than conditioning on the entire history $\boldsymbol{x}_{\leq t}$. Recalling Eq. 8, whether or not $\widehat{F}(r \mid \boldsymbol{h}_t)$ is a good approximation of $F(r \mid \boldsymbol{x}_{\leq t})$ entirely depends on the ability of the ESN to encode all relevant information from the past in its state. In particular, if the state representation captures all the relevant information, then RESCP provides asymptotic conditional coverage given such a history. Conversely, in the extreme case where every sequence is mapped by the reservoir into the same uninformative state, RESCP would simply provide marginal coverage, as one would expect. This introduces an additional bias-variance tradeoff. For example, the reservoir mapping all trajectories to similar states would result in small variance, but would have small discriminative power (large bias). We refer the reader to the rich body of literature on the expressiveness of ESNs, particularly regarding their universal approximation properties (Grigoryeva & Ortega, 2018a;b; Li & Yang, 2025) and their effectiveness in extracting meaningful representations from time series data (Bianchi et al., 2020). In particular, we refer to (Lukoševičius, 2012) for practical guidelines on how to set up ESNs effectively.

### 3.1.2 TIME-DEPENDENT WEIGHTS

RESCP, by default, compares the query state with *any* time step from the calibration set. Note that state representations do not include any positional encoding that accounts for how far back in time a certain sample is, treating recent and distant residuals alike. This is not problematic for time-invariant and stable processes, which we assumed in 3.1.1 to provide coverage guarantees. To deal, instead, with distribution shifts, we 1) update the calibration set over time, keeping its size $N$ fixed using a first-in-first-out approach, and 2) make the weights time-dependent, following an approach similar to NexCP (Barber et al., 2023). In particular, we condition the weights on the distance between time steps as

$$w_i(\boldsymbol{h}_t, t) = \gamma(\Delta(t, i)) w_i(\boldsymbol{h}_t), \tag{14}$$

where $\gamma : \mathbb{N} \to \mathbb{R}$ is a discount function that maps the distance between time steps $\Delta(t, i)$ to a decay factor. The discount schedule can be chosen in different ways, e.g., exponential like NexCP (Barber et al., 2023) or linear. In our settings, we observed that a linear decay–i.e., $\gamma(\Delta(t, i)) := 1/\Delta(t, i)$– allowed to keep PIs up-to-date without reducing too much the effective sample size. This is particularly important if assumption 3.1 is violated and the underlying dynamics change over time. It is worth noting that RESCP is a non-parametric approach without learnable parameters that is much more robust to non-stationarity compared to approaches that train a model (Auer et al., 2023; Cini et al., 2025), which must necessarily be updated when time-invariance is lost.

## 3.2 RESERVOIR CONFORMAL QUANTILE REGRESSION

Usually CP methods operate directly on conformal scores. However, in some case it might be beneficial to consider exogenous inputs (covariates). Adding exogenous variables as input to the reservoir can affect the internal dynamics of the network and, thus, its states. Depending on the characteristics of specific exogenous variables, they can harm the effectiveness of the reservoir in localizing prediction w.r.t. the dynamics of the target variable. Since the network is not trained, adjusting to the relevance and characteristics of the covariates is not possible with the completely unsupervised approach discussed in Sec. 3.1. As an alternative to RESCP in these scenarios, we consider a variant of our original approach, called Reservoir Conformal Quantile Regression (RESCQR), which can account for exogenous inputs by relying on quantile regression. In practice, we use a linear readout to map the state at each time step to a set of quantiles of interest. Specifically, we use states $\boldsymbol{h}_{1:T-H}$ and residuals $r_{H+1:T}$ in the calibration set to fit a linear model $\widehat{Q}_\alpha(\boldsymbol{h}_t)$ as a quantile predictor. To do so, we train the readout to minimize the pinball loss for the target quantiles $\{\beta_1, \ldots, \beta_M\}$ of the conformity scores:

$$\mathcal{L}_{\beta_i}(\widehat{Q}_{\beta_i}(\boldsymbol{h}_t), r) = \begin{cases} (1 - \beta_i)(\widehat{Q}_{\beta_i}(\boldsymbol{h}_t) - r), & \widehat{Q}_{\beta_i}(\boldsymbol{h}_t) \geq r \\ \beta_i(r - \widehat{Q}_{\beta_i}(\boldsymbol{h}_t)), & \widehat{Q}_{\beta_i}(\boldsymbol{h}_t) < r \end{cases}$$

We can then construct PIs as discussed in Sec. 3.1. As we will show in Sec. 4, RESCQR provides a practical methods approach that can work well in scenarios where the availability of enough calibration data allows for training the readout and informative exogenous variables are available.

Table 1: Performance comparison for $\alpha = 0.1$. $\Delta$Cov values are color-coded for undercoverage cases: yellow (1-2%), orange (2-4%), red (>4%). The top three Winkler scores for each scenario are highlighted: **bold+underlined** (1st), underlined (2nd), dot-underlined (3rd).

| | | Metric | *Learning* | | | | *Non-learning* | | |
|---|---|---|---|---|---|---|---|---|---|
| | | | SPCI | HopCPT | CP-QRNN | **RESCQR** | SCP | NexCP | **RESCP** |
| Solar | RNN | $\Delta$Cov | $0.05_{\pm0.17}$ | $-1.64_{\pm1.18}$ | $-0.26_{\pm0.92}$ | $-1.10_{\pm0.91}$ | $0.37$ | $1.46$ | $0.74_{\pm0.24}$ |
| | | PI-Width | $70.41_{\pm1.73}$ | $60.49_{\pm2.10}$ | $55.74_{\pm0.98}$ | $59.99_{\pm1.72}$ | $79.15$ | $89.56$ | $62.25_{\pm0.75}$ |
| | | Winkler | $151.50_{\pm1.18}$ | $112.46_{\pm9.34}$ | **$78.42_{\pm0.20}$** | $82.76_{\pm0.26}$ | $171.41$ | $164.96$ | $104.24_{\pm0.79}$ |
| | Transf | $\Delta$Cov | $-0.16_{\pm0.46}$ | $1.32_{\pm0.62}$ | $1.37_{\pm2.09}$ | $-3.51_{\pm16.26}$ | $0.35$ | $1.46$ | $3.09_{\pm0.35}$ |
| | | PI-Width | $71.36_{\pm3.21}$ | $61.49_{\pm1.74}$ | $55.70_{\pm0.95}$ | $59.56_{\pm1.59}$ | $79.03$ | $89.21$ | $63.34_{\pm1.11}$ |
| | | Winkler | $152.85_{\pm0.67}$ | $107.59_{\pm6.07}$ | **$77.61_{\pm0.25}$** | $82.16_{\pm0.32}$ | $169.64$ | $163.34$ | $103.13_{\pm0.58}$ |
| | ARIMA | $\Delta$Cov | $0.51_{\pm0.36}$ | $2.88_{\pm2.13}$ | $-0.41_{\pm0.42}$ | $-2.03_{\pm0.62}$ | $0.16$ | $1.76$ | $0.68_{\pm0.95}$ |
| | | PI-Width | $91.71_{\pm1.18}$ | $143.32_{\pm7.36}$ | $59.17_{\pm0.41}$ | $66.19_{\pm0.81}$ | $124.19$ | $137.53$ | $77.17_{\pm2.07}$ |
| | | Winkler | $148.86_{\pm0.34}$ | $173.49_{\pm5.15}$ | **$77.34_{\pm0.30}$** | $85.38_{\pm0.45}$ | $215.77$ | $207.58$ | $110.38_{\pm4.03}$ |
| Beijing | RNN | $\Delta$Cov | $-1.73_{\pm0.67}$ | $-5.18_{\pm12.67}$ | $-1.86_{\pm2.16}$ | $-1.21_{\pm1.65}$ | $-0.32$ | $0.00$ | $-0.70_{\pm0.77}$ |
| | | PI-Width | $67.93_{\pm1.43}$ | $68.47_{\pm13.30}$ | $61.71_{\pm4.91}$ | $65.53_{\pm4.09}$ | $67.99$ | $69.79$ | $65.96_{\pm2.50}$ |
| | | Winkler | $124.51_{\pm2.25}$ | $140.50_{\pm43.64}$ | **$104.03_{\pm0.99}$** | $105.43_{\pm0.85}$ | $126.41$ | $124.10$ | $106.07_{\pm0.47}$ |
| | Transf | $\Delta$Cov | $-0.98_{\pm1.04}$ | $-8.05_{\pm16.41}$ | $-1.07_{\pm0.69}$ | $-1.43_{\pm1.10}$ | $-0.41$ | $0.03$ | $-0.49_{\pm0.59}$ |
| | | PI-Width | $69.96_{\pm4.53}$ | $61.76_{\pm14.39}$ | $62.41_{\pm1.66}$ | $64.41_{\pm2.72}$ | $67.46$ | $69.64$ | $64.06_{\pm1.74}$ |
| | | Winkler | $125.41_{\pm1.43}$ | $140.30_{\pm36.01}$ | **$102.81_{\pm0.44}$** | $105.97_{\pm1.21}$ | $126.63$ | $124.35$ | $103.64_{\pm0.21}$ |
| | ARIMA | $\Delta$Cov | $-0.23_{\pm0.40}$ | $-1.37_{\pm0.26}$ | $-1.54_{\pm0.77}$ | $-1.42_{\pm1.15}$ | $-0.24$ | $-0.16$ | $0.63_{\pm0.22}$ |
| | | PI-Width | $74.68_{\pm1.21}$ | $67.78_{\pm0.50}$ | $61.80_{\pm1.68}$ | $66.01_{\pm3.01}$ | $75.72$ | $76.45$ | $70.43_{\pm0.86}$ |
| | | Winkler | $130.59_{\pm0.59}$ | $122.48_{\pm4.36}$ | **$101.84_{\pm0.67}$** | $107.20_{\pm1.21}$ | $135.07$ | $132.03$ | $108.75_{\pm0.31}$ |
| Exchange | RNN | $\Delta$Cov | $2.98_{\pm0.65}$ | $2.75_{\pm0.08}$ | $-1.07_{\pm2.52}$ | $3.18_{\pm1.25}$ | $2.29$ | $1.64$ | $1.13_{\pm0.27}$ |
| | | PI-Width | $0.0241_{\pm0.0007}$ | $0.0404_{\pm0.0001}$ | $0.0341_{\pm0.0018}$ | $0.0383_{\pm0.0008}$ | $0.0444$ | $0.0405$ | $0.0210_{\pm0.0001}$ |
| | | Winkler | $0.0287_{\pm0.0007}$ | $0.0482_{\pm0.0001}$ | $0.0461_{\pm0.0005}$ | $0.0464_{\pm0.0005}$ | $0.0517$ | $0.0492$ | **$0.0264_{\pm0.0002}$** |
| | Transf | $\Delta$Cov | $4.44_{\pm0.35}$ | $2.98_{\pm0.07}$ | $-0.57_{\pm1.58}$ | $0.82_{\pm1.89}$ | $4.57$ | $3.25$ | $1.46_{\pm0.18}$ |
| | | PI-Width | $0.0255_{\pm0.0005}$ | $0.0399_{\pm0.0001}$ | $0.0337_{\pm0.0016}$ | $0.0365_{\pm0.0013}$ | $0.0544$ | $0.0509$ | $0.0229_{\pm0.0001}$ |
| | | Winkler | $0.0300_{\pm0.0005}$ | $0.0479_{\pm0.0001}$ | $0.0480_{\pm0.0009}$ | $0.0475_{\pm0.0008}$ | $0.0620$ | $0.0602$ | **$0.0294_{\pm0.0001}$** |
| | ARIMA | $\Delta$Cov | $3.49_{\pm0.41}$ | $2.07_{\pm0.08}$ | $-1.22_{\pm1.78}$ | $0.68_{\pm1.58}$ | $3.08$ | $2.13$ | $0.38_{\pm0.41}$ |
| | | PI-Width | $0.0242_{\pm0.0006}$ | $0.0379_{\pm0.0000}$ | $0.0330_{\pm0.0007}$ | $0.0351_{\pm0.0009}$ | $0.0387$ | $0.0356$ | $0.0207_{\pm0.0001}$ |
| | | Winkler | $0.0289_{\pm0.0003}$ | $0.0456_{\pm0.0001}$ | $0.0455_{\pm0.0003}$ | $0.0455_{\pm0.0006}$ | $0.0462$ | $0.0447$ | **$0.0268_{\pm0.0001}$** |
| ACEA | RNN | $\Delta$Cov | $-0.78_{\pm1.88}$ | $-2.18_{\pm0.00}$ | $-12.37_{\pm8.98}$ | $-18.86_{\pm7.44}$ | $-0.99$ | $-0.33$ | $1.56_{\pm0.62}$ |
| | | PI-Width | $8.99_{\pm0.68}$ | $18.90_{\pm0.00}$ | $15.86_{\pm1.99}$ | $15.23_{\pm1.96}$ | $19.63$ | $20.15$ | $9.61_{\pm0.26}$ |
| | | Winkler | $14.27_{\pm0.19}$ | $27.56_{\pm0.00}$ | $32.61_{\pm5.69}$ | $34.61_{\pm3.53}$ | $27.60$ | $26.83$ | **$12.91_{\pm0.23}$** |
| | Transf | $\Delta$Cov | $-1.41_{\pm1.29}$ | $-2.51_{\pm0.00}$ | $-13.35_{\pm9.85}$ | $-26.92_{\pm7.68}$ | $-5.52$ | $-0.45$ | $3.54_{\pm0.32}$ |
| | | PI-Width | $9.10_{\pm0.23}$ | $18.29_{\pm0.00}$ | $14.82_{\pm2.02}$ | $13.20_{\pm1.47}$ | $16.53$ | $20.20$ | $10.10_{\pm0.16}$ |
| | | Winkler | $14.58_{\pm0.36}$ | $27.53_{\pm0.00}$ | $33.47_{\pm7.18}$ | $39.98_{\pm4.27}$ | $29.24$ | $27.47$ | **$12.90_{\pm0.16}$** |
| | ARIMA | $\Delta$Cov | $1.41_{\pm0.90}$ | $-3.58_{\pm0.00}$ | $-29.35_{\pm11.01}$ | $-27.10_{\pm8.65}$ | $-0.75$ | $-0.40$ | $5.02_{\pm0.40}$ |
| | | PI-Width | $12.46_{\pm0.35}$ | $34.84_{\pm0.00}$ | $18.16_{\pm2.43}$ | $17.39_{\pm2.20}$ | $38.13$ | $36.08$ | $13.63_{\pm0.55}$ |
| | | Winkler | $17.36_{\pm0.13}$ | $44.69_{\pm0.00}$ | $53.89_{\pm9.33}$ | $48.49_{\pm6.89}$ | $45.99$ | $43.70$ | **$16.21_{\pm0.53}$** |

# 4 EXPERIMENTS

We compare the performances of RESCP against state-of-the-art conformal prediction baselines on time series data coming from several applications.

**Datasets and baselines** We evaluated our method across four datasets. 1) The **Solar** dataset comes from the US National Solar Radiation Database (Sengupta et al., 2018). We used the dataset containing 50 time series from different locations over a period of 3 years, as done in previous work (Auer et al., 2023). 2) The **Beijing** dataset contains air quality measurements taken over a period of 4 years from 12 locations in the city of Beijing, China (Zhang et al., 2017). 3) The **Exchange** dataset consists of a collection of the daily exchange rates of eight countries from 1990 to 2016 (Lai et al., 2017). 4) **ACEA** contains electricity consumption data coming from the backbone of the energy supply network in the city of Rome (Bianchi et al., 2015). More details are reported in App. B. We compare RESCP and its RESCQR variant against the competitors presented in Sec. 2: 1) vanilla

Table 2: Runtime (in seconds) using the RNN point forecasting baseline.

| Dataset | Learning | | | | Non-learning | | |
|---|---|---|---|---|---|---|---|
| | SPCI | HopCPT | CP-QRNN | **RESCQR** | SCP | NexCP | **RESCP** |
| Solar | $1039.8_{\pm 2}$ | $4574.6_{\pm 1356.5}$ | $172.4_{\pm 17}$ | $82.1_{\pm 6.7}$ | 18 | 66 | $52.9_{\pm 14.7}$ |
| Beijing | $351.4_{\pm 2.1}$ | $1838.5_{\pm 202.4}$ | $81.6_{\pm 3}$ | $46_{\pm 2.1}$ | 9 | 29 | $34.6_{\pm 1.4}$ |
| Exchange | $50.6_{\pm 0.5}$ | $318_{\pm 0}$ | $36.9_{\pm 1.2}$ | $16_{\pm 0.9}$ | 2 | 2 | $6.5_{\pm 0.5}$ |
| ACEA | $227.6_{\pm 5.8}$ | $2262.8_{\pm 918.9}$ | $95.3_{\pm 9.5}$ | $56.6_{\pm 2.9}$ | 7 | 57 | $70.5_{\pm 0.5}$ |

**SCP** (Vovk et al., 2005), 2) **NexCP** (Barber et al., 2023), 3) **SPCI** (Xu & Xie, 2023a), and 4) **HopCPT** (Auer et al., 2023). We also include 5) **CP-QRNN**, an RNN with a multi-dimensional output trained to perform quantile regression on the calibration set using the pinball loss. The architecture is analogous to the model called CORNN introduced in Cini et al. (2025) (more details in App. C). For RESCP, we use time-dependent weights (with a linear decay schedule) and the cosine similarity between reservoir states as the similarity score.

**Experimental setup and evaluation metrics** In our experiments, we adopted a $40\%/40\%/20\%$ split for training, calibration, and testing sets, respectively. As base models, we consider three different point predictors: a simple RNN with gated recurrent cells (Cho et al., 2014), a decoder-only Transformer (Vaswani et al., 2017), and an ARIMA model (Box & Jenkins, 1970). After training, we evaluated each model on the calibration set and saved the residuals, which we then used in all the baselines to compute the PIs. As evaluation metrics, we considered the $\Delta$Cov, i.e., the difference between the specified confidence level $1 - \alpha$ and the achieved coverage on the test set, the width of the PIs, and the Winkler score (Winkler, 1972), which penalizes the PI width whenever the observed value falls outside the computed interval, with the penalty scaled proportionally to the magnitude of the deviation. Model selection for all methods is done by minimizing the Winkler score over a validation set, except for HopCPT, which follows a custom procedure (Auer et al., 2023). More details on model selection for our methods are reported in App. C. For RESCP, when approximating the quantiles via Monte Carlo sampling, we set the number of observations used for calibration as the sample size. As mentioned in Sec. 3.1, we account for skewed distributions by building PIs as in Eq. 12. The optimal $\beta$ is chosen from 100 linearly spaced values between 0 to $\alpha$.

## 4.1 RESULTS

Results across the datasets and base models are summarized in Tab. 1, where CP methods are grouped based on whether they rely on learning or not. Additional results for different miscoverage levels can be found in App. D (see Tab. 5 and Tab. 6). RESCP achieves competitive performance across all settings and datasets, both in terms of coverage and Winkler score, while remaining highly scalable. Notably, RESCPs outperforms HopCPT in almost all scenarios. This is particularly remarkable since both methods use weighted empirical distributions to model uncertainty, but HopCPT requires training an attention-based architecture end-to-end, while RESCPs does not perform any form of training. Moreover, similarly to NexCP and differently from most of the other methods, RESCP provides approximately valid coverage across all scenarios, but with large improvements in PI width, which is reduced up to 60%. The CP-QRNN baseline achieves strong

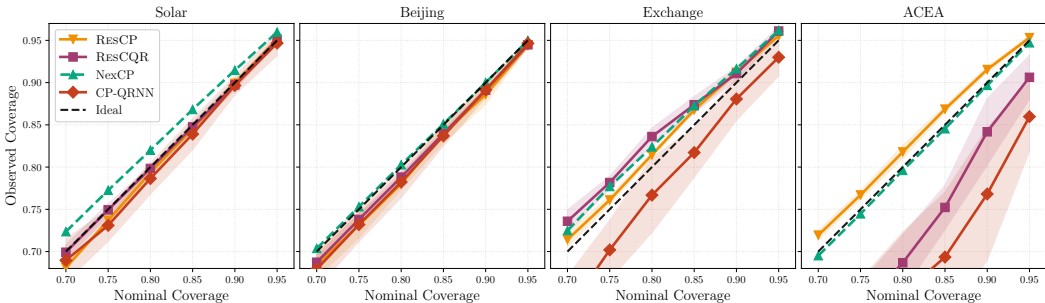

Figure 2: Calibration curves of RESCP, RESCQR, CP-QRNN and NexCP over all datasets with the RNN baseline.

Table 3: Ablation of **RESCP**, RESCP without decay, RESCP without sliding window, and the combination of the two, using the RNN base model. Best Winkler scores are shown in bold.

| | Metric | **RESCP** | No decay | No window | No window, no decay |
|---|---|---|---|---|---|
| Solar | $\Delta$Cov | $0.74_{\pm0.24}$ | $-0.10_{\pm0.26}$ | $0.89_{\pm0.20}$ | $1.70_{\pm0.10}$ |
| | PI-Width | $62.25_{\pm0.75}$ | $60.59_{\pm0.84}$ | $59.81_{\pm0.87}$ | $60.90_{\pm0.30}$ |
| | Winkler | $\mathbf{104.24_{\pm0.79}}$ | $107.46_{\pm0.49}$ | $104.70_{\pm0.20}$ | $104.46_{\pm0.40}$ |
| Beijing | $\Delta$Cov | $-0.70_{\pm0.77}$ | $-1.64_{\pm0.61}$ | $0.10_{\pm1.36}$ | $1.64_{\pm0.88}$ |
| | PI-Width | $65.96_{\pm2.50}$ | $64.04_{\pm1.69}$ | $64.91_{\pm3.36}$ | $69.53_{\pm3.02}$ |
| | Winkler | $106.07_{\pm0.47}$ | $114.85_{\pm0.29}$ | $\mathbf{97.99_{\pm0.31}}$ | $98.25_{\pm0.66}$ |
| Exch. | $\Delta$Cov | $1.13_{\pm0.27}$ | $2.01_{\pm0.19}$ | $4.19_{\pm0.05}$ | $4.20_{\pm0.12}$ |
| | PI-Width | $0.0210_{\pm0.0001}$ | $0.0219_{\pm0.0001}$ | $0.0249_{\pm0.0002}$ | $0.0254_{\pm0.0003}$ |
| | Winkler | $\mathbf{0.0264_{\pm0.0002}}$ | $0.0269_{\pm0.0002}$ | $0.0284_{\pm0.0001}$ | $0.0291_{\pm0.0002}$ |
| ACEA | $\Delta$Cov | $1.56_{\pm0.62}$ | $2.79_{\pm0.45}$ | $5.34_{\pm0.38}$ | $4.96_{\pm0.63}$ |
| | PI-Width | $9.61_{\pm0.26}$ | $10.15_{\pm0.09}$ | $11.88_{\pm0.09}$ | $12.15_{\pm0.17}$ |
| | Winkler | $\mathbf{12.91_{\pm0.23}}$ | $13.41_{\pm0.08}$ | $14.80_{\pm0.17}$ | $15.25_{\pm0.40}$ |

performance on the large datasets with informative exogenous variables, outperforming all methods in Solar, but fails in achieving good results in the smaller datasets, such as ACEA and Exchange. Moreover, on those datasets, trainable methods obtain worse performance, likely due to distribution changes that require the trained models to be updated over time. RESCQR achieves competitive performance against the competitors, including more complex models such as CP-QRNN in Solar, while being more scalable. All other baselines obtain good coverage in most settings, but produce prediction intervals that are much more conservative. The runtime of each CP method is reported in Tab. 2 and shows the advantage of RESCP in terms of scalability against methods that require fitting a model. Moreover, as RESCP does not need centralized training on a GPU: it can be easily scaled in a distributed setting.

**Additional experiments and ablation studies**  Fig. 2 presents calibration curves for RESCP and RESCQR and the CP-QRNN and NexCP baselines, across all datasets and using an RNN as the base point predictor. The curves show that RESCP provides accurate estimates at all the considered coverage levels. Note that while NexCP is well calibrated, it produces much wider intervals (see Tab. 1, Tab. 5 and Tab. 6). Additional results are provided in App. D, including an experiment of RESCP on a non-stationary synthetic dataset. In Tab. 3, we also report an ablation study of RESCP. In particular, we evaluated the impact of 1) removing time-dependent weights (**No decay**), 2) using all the available samples for calibration rather than only the most recent ones (**No window**), and 3) the combination of the previous two ablations (**No window, no decay**). The results clearly show the impcat of the proposed designs. More comprehensive ablation results for all base models are presented in App. E, which also include an ablation study on the use of exogenous variables. Finally, App. F reports additional results on the sensitivity of RESCP to different hyperparameter configurations.

## 5  CONCLUSION AND FUTURE WORKS

We introduced *Reservoir Conformal Prediction* (RESCP), a simple, fast, and effective method for uncertainty quantification in time series forecasting that provides accurate uncertainty estimation across several tasks. We showed that RESCP is principled and theoretically sound. In particular, assuming strong mixing conditions and reasonable regularity conditions, we demonstrated that RESCP provides asymptotically valid conditional coverage guarantees. Results on a diverse suite of benchmarks show that RESCP achieves state-of-the-art performance while being more scalable and computationally efficient than its competitors. There are several directions for future work. In particular, it would be interesting to explore extensions to modeling joint distributions in multistep forecasting (Sun & Yu, 2024), treat multidimensional time series (Feldman et al., 2023; Xu et al., 2024), and spatiotemporal data (Cini et al., 2025).

ACKNOWLEDGMENTS

This work is supported by the Research Council of Norway through *RELAY: Relational Deep Learning for Energy Analytics* (project no. 345017) and through its Centre of Excellence *Integreat - The Norwegian Centre for knowledge-driven machine learning* (project no. 332645), the Swiss National Science Foundation grant no. 225351 (*Relational Deep Learning for Reliable Time Series Forecasting at Scale*), the EPSRC Turing AI World-Leading Research Fellowship No. EP/X040062/1 and EPSRC AI Hub No. EP/Y028872/1. The authors wish to thank Nvidia Corporation for donating some of the GPUs used in this project and Carlo Abate for the help in checking the proof.

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

# APPENDIX

## A  PROOFS FOR THE ASYMPTOTIC CONDITIONAL COVERAGE

We start the proof with the following lemma.

**Lemma A.1** (Regularity of the conditional CDF). *Under Assumptions 3.2–3.3, the conditional CDF $F(r \mid \boldsymbol{h}_t)$ is continuous in $\boldsymbol{h}_t \in \mathbb{R}^{d_h}$, uniformly in $r \in \mathbb{R}$.*

*Proof.* By Assumption 3.2, the map $\boldsymbol{x}_{\leq t} \mapsto \boldsymbol{h}_t$ is Lipschitz with respect to $d_{\mathrm{fm}}$. By Assumption 3.3, the map $\boldsymbol{x}_{\leq t} \mapsto F(r \mid \boldsymbol{x}_{\leq t})$ is continuous. Therefore the composition $\boldsymbol{x}_{\leq t} \mapsto \boldsymbol{h}_t \mapsto F(r \mid \boldsymbol{h}_t)$ is continuous in $\boldsymbol{h}_t$. □

We then continue with the proof of Theorem 3.6, which we start by proving point wise convergence.

*Proof of Theorem 3.6.* Define

$$\bar{F}_n(r \mid \boldsymbol{h}_t) := \sum_{i=1}^{n} w_i^{(n)}(\boldsymbol{h}_t) F(r \mid \boldsymbol{h}_i).$$

Decompose

$$\widehat{F}_n(r \mid \boldsymbol{h}_t) - F(r \mid \boldsymbol{h}_t) = \underbrace{(\widehat{F}_n(r \mid \boldsymbol{h}_t) - \bar{F}_n(r \mid \boldsymbol{h}_t))}_{\text{(I)}} + \underbrace{(\bar{F}_n(r \mid \boldsymbol{h}_t) - F(r \mid \boldsymbol{h}_t))}_{\text{(II)}}.$$

We handle the two terms separately and show both going uniformly to 0 in probability.

**Term (I):**

$$\widehat{F}_n(r \mid \boldsymbol{h}_t) - \bar{F}_n(r \mid \boldsymbol{h}_t) = \sum_{i=1}^{n} w_i^{(n)}(\boldsymbol{h}_t)\big(\mathbb{1}\{r_{i+H} \leq r\} - F(r \mid \boldsymbol{h}_i)\big) = \sum_{i=1}^{n} w_i^{(n)}(\boldsymbol{h}_t) Y_i(r) = S_n(r)$$

with $Y_i(r) = \mathbb{1}\{r_{i+H} \leq r\} - F(r \mid \boldsymbol{h}_i)$. We can see that $\mathbb{E}\left[Y_i(r) \mid \boldsymbol{h}_i\right] = 0$, since

$$\mathbb{E}[\mathbb{1}\{r_i \leq r\} \mid \boldsymbol{h}_i] = 1 \cdot \mathbb{P}(r_i \leq r \mid \boldsymbol{h}_i) + 0 \cdot \mathbb{P}(r_i > r \mid \boldsymbol{h}_i) = \mathbb{P}(r_i \leq r \mid \boldsymbol{h}_i) =: F(r \mid \boldsymbol{h}_i).$$

Therefore, $\mathbb{E}\left[S_n(r)\right] = 0$. Moreover, $|Y_i(r)| \leq 1$. To prove that $S_n(r) \xrightarrow[n \to \infty]{\mathbb{P}} 0$, we leverage Chebyshev's inequality

$$\mathbb{P}(|S_n(r)| > \epsilon) \leq \frac{\mathrm{Var}(S_n(r))}{\epsilon^2}.$$

and then expand the variance as

$$\mathrm{Var}(S_n(r)) = \mathrm{Var}\left(\sum_{i=1}^{n} w_i^{(n)} Y_i\right) = \sum_{i=1}^{n} (w_i^{(n)})^2 \mathrm{Var}(Y_i) + \sum_{i \neq j} w_i^{(n)} w_j^{(n)} \mathrm{Cov}(Y_i, Y_j) \quad (15)$$

$$\leq \sum_{i=1}^{n} (w_i^{(n)})^2 \mathrm{Var}(Y_i) + \sum_{i \neq j} w_i^{(n)} w_j^{(n)} |\mathrm{Cov}(Y_i, Y_j)|.$$

Starting by the variances term, since $|Y_i| \leq 1$, we have $\mathrm{Var}(Y_i) \leq 1$, thus the first term of Eq. 15 is bounded $\sum_{i=1}^{n} (w_i^{(n)})^2 \mathrm{Var}(Y_i) \leq \sum_{i=1}^{n} (w_i^{(n)})^2 \to 0$ thanks to Assumption 3.5 (i). We now move to the covariances. From Lemma 3 of Doukhan (1994), we know that since the process generating $(\boldsymbol{x}_i, r_{i+H})$ is mixing by Assumption 3.1, for bounded variables $Y_i, Y_j$ that are function of this process the following holds

$$|\mathrm{Cov}(Y_i, Y_j)| \leq 4\alpha(|i - j|),$$

where $\alpha(k)$ is the mixing coefficient. Hence, in our case,

$$\sum_{i \neq j} w_i^{(n)} w_j^{(n)} |\mathrm{Cov}(Y_i, Y_j)| \leq 4 \sum_{i \neq j} w_i^{(n)} w_j^{(n)} \alpha(|i - j|).$$

We can split the sum in two terms

$$4 \underbrace{\sum_{1 \leq |i-j| \leq K} w_i^{(n)} w_j^{(n)} \alpha(|i-j|)}_{(*)} + 4 \underbrace{\sum_{|i-j| > K} w_i^{(n)} w_j^{(n)} \alpha(|i-j|)}_{(**)}$$

such that for all lags $k > K$ the mixing coefficient is arbitrary small, i.e., for any $\epsilon > 0, \alpha(k) < \epsilon$ (this is always possible).

$$(*) \leq \left( \max_{1 \leq k \leq K} \alpha(k) \right) \sum_{1 \leq |i-j| \leq K} w_i^{(n)} w_j^{(n)}$$

$$\leq A_{max} \sum_{1 \leq |i-j| \leq K} w_i^{(n)} w_j^{(n)}$$

$$= 2 A_{max} \sum_{k=1}^{K} \sum_{i=1}^{n-k} w_i^{(n)} w_{i+k}^{(n)}$$

Using the inequality $2ab \leq a^2 + b^2$:

$$2 A_{max} \sum_{k=1}^{K} \sum_{i=1}^{n-k} w_i^{(n)} w_{i+k}^{(n)} \leq A_{max} \sum_{k=1}^{K} \left( \sum_{i=1}^{n-k} (w_i^{(n)})^2 + \sum_{i=1}^{n-k} (w_{i+k}^{(n)})^2 \right)$$

$$\leq 2 A_{max} \sum_{k=1}^{K} \sum_{i=1}^{n} (w_i^{(n)})^2$$

$$= 2 A_{max} K \sum_{i=1}^{n} (w_i^{(n)})^2$$

$$= \frac{2 A_{max} K}{m_n} \to 0.$$

For the second term

$$(**) = \sum_{|i-j| > K} w_i^{(n)} w_j^{(n)} \alpha(|i-j|)$$

$$\leq \epsilon \sum_{|i-j| > K} w_i^{(n)} w_j^{(n)}$$

$$\leq \epsilon$$

where $\epsilon$ is arbitrarily small.

**Term (II):**

$$\bar{F}_n(r \mid \boldsymbol{h}_t) - F(r \mid \boldsymbol{h}_t) = \sum_{i=1}^{n} w_i^{(n)}(\boldsymbol{h}_t) \big( F(r \mid \boldsymbol{h}_i) - F(r \mid \boldsymbol{h}_t) \big)$$

Because weights are a convex combination (nonnegative, sum to 1),

$$|\bar{F}_n(r \mid \boldsymbol{h}_t) - F(r \mid \boldsymbol{h}_t)| \leq \sum_{i=1}^{n} w_i^{(n)}(\boldsymbol{h}_t) \big| F(r \mid \boldsymbol{h}_i) - F(r \mid \boldsymbol{h}_t) \big|$$

Now we use Lemma A.1. For any $\epsilon > 0$ we can choose $\delta > 0$ so that $\|\boldsymbol{h}_i - \boldsymbol{h}_t\| < \delta$ implies $\sup_b \big| F(r \mid \boldsymbol{h}_i) - F(r \mid \boldsymbol{h}_t) \big| < \epsilon$. We can split the sum into indices with $\|\boldsymbol{h}_i - \boldsymbol{h}_t\| < \delta$ and $\|\boldsymbol{h}_i - \boldsymbol{h}_t\| \geq \delta$:

$$\sum_{i=1}^{n} w_i^{(n)}(\boldsymbol{h}_t) \big| F(r \mid \boldsymbol{h}_i) - F(r \mid \boldsymbol{h}_t) \big| = \sum_{i: \|\boldsymbol{h}_i - \boldsymbol{h}_t\| < \delta} w_i^{(n)}(\boldsymbol{h}_t) \big| F(r \mid \boldsymbol{h}_i) - F(r \mid \boldsymbol{h}_t) \big|$$

$$+ \sum_{i: \|\boldsymbol{h}_i - \boldsymbol{h}_t\| \geq \delta} w_i^{(n)}(\boldsymbol{h}_t) \big| F(r \mid \boldsymbol{h}_i) - F(r \mid \boldsymbol{h}_t) \big|$$

The first term is smaller than $\epsilon$,

$$\sum_{i:\|\boldsymbol{h}_i-\boldsymbol{h}_t\|<\delta} w_i^{(n)}(\boldsymbol{h}_t)\big|F(r\mid\boldsymbol{h}_i)-F(r\mid\boldsymbol{h}_t)\big| < \epsilon \sum_{i:\|\boldsymbol{h}_i-\boldsymbol{h}_t\|<\delta} w_i^{(n)}(\boldsymbol{h}_t) < \epsilon$$

which we can make arbitrarily small. The second term goes to $0$ in probability under Assumptions 3.5-(i) and 3.5-(ii): as the temperature goes to zero, the weight concentrates on observations $\boldsymbol{h}_i$ close to $\boldsymbol{h}_t$, and the weight of points not in the neighborhood of $\boldsymbol{h}_t$ vanishes

$$\sum_{i:\|\boldsymbol{h}_i-\boldsymbol{h}_t\|\geq\delta} w_i^{(n)}(\boldsymbol{h}_t)\big|F(r\mid\boldsymbol{h}_i)-F(r\mid\boldsymbol{h}_t)\big| \leq 2\sum_{i:\|\boldsymbol{h}_i-\boldsymbol{h}_t\|\geq\delta} w_i^{(n)}(\boldsymbol{h}_t) \xrightarrow[n\to\infty]{\mathbb{P}} 0.$$

**Uniform convergence:** The proof here is analogous to proofs of the Glivenko–Cantelli theorem. Consider points

$$-\infty = r_0 < r_1 < ... < r_{K-1} < r_K = \infty$$

such that for all $\epsilon > 0$

$$F(r_k\mid\boldsymbol{h}_t) - F(r_{k-1}\mid\boldsymbol{h}_t) \leq \epsilon.$$

This implies

$$\sup_{r\in\mathbb{R}}\big|\widehat{F}_n(r\mid\boldsymbol{h}_t)-F(r\mid\boldsymbol{h}_t)\big| \leq \max_{1\leq k\leq K}\big|\widehat{F}_n(r_k\mid\boldsymbol{h}_t)-F(r_k\mid\boldsymbol{h}_t)\big| + \epsilon$$

Since we already proven pointwise convergence, we have that

$$\max_{1\leq k\leq K}\big|\widehat{F}_n(r_k\mid\boldsymbol{h}_t)-F(r_k\mid\boldsymbol{h}_t)\big| \xrightarrow[n\to\infty]{\mathbb{P}} 0$$

and since $\epsilon$ can be made arbitrary small

$$\sup_{r\in\mathbb{R}}\big|\widehat{F}_n(r\mid\boldsymbol{h}_t)-F(r\mid\boldsymbol{h}_t)\big| \xrightarrow[n\to\infty]{\mathbb{P}} 0.$$

$\square$

*Proof of Corollary 3.7.* Starting from

$$\mathbb{P}\left(y_{t+H}\in\hat{C}_t^\alpha(\hat{y}_{t+H})\mid\boldsymbol{h}_t\right) = \mathbb{P}\left(r_{t+H}\in\left[\widehat{Q}_{\alpha/2}(\boldsymbol{h}_t),\widehat{Q}_{1-\alpha/2}(\boldsymbol{h}_t)\right]\mid\boldsymbol{h}_t\right)$$

so we have

$$\mathbb{P}\left(r_{t+H}\in\left[\widehat{Q}_{\alpha/2}(\boldsymbol{h}_t),\widehat{Q}_{1-\alpha/2}(\boldsymbol{h}_t)\right]\mid\boldsymbol{h}_t\right) = F\left(\widehat{Q}_{1-\alpha/2}(\boldsymbol{h}_t)\big|\boldsymbol{h}_t\right) - F\left(\widehat{Q}_{\alpha/2}(\boldsymbol{h}_t)\big|\boldsymbol{h}_t\right)$$

and since Theorem 3.6 gives us the consistency of the quantile estimator to the true conditional quantile, we have that

$$F\left(\widehat{Q}_{1-\alpha/2}(\boldsymbol{h}_t)\big|\boldsymbol{h}_t\right) - F\left(\widehat{Q}_{\alpha/2}(\boldsymbol{h}_t)\big|\boldsymbol{h}_t\right) \xrightarrow[n\to\infty]{\mathbb{P}} 1-\alpha.$$

$\square$

# B  DATASETS AND POINT PREDICTORS

## B.1  DATASETS

In the experiments that we performed, we considered four datasets coming from different real-world scenarios and application domains.

**Solar data**    The solar data comes from the US National Solar Radiation Database (Sengupta et al., 2018). We worked with the dataset of 50 time series generated by Auer et al. (2023), each consisting of 52,609 half-hourly measurements. These were resampled to obtain 26,304 hourly observations. Solar radiation (`dhi`, Diffuse Horizontal Irradiance) was designated as the target variable, and the dataset includes eight additional environmental features:

- `dni`, Direct Normal Irradiance: amount of solar radiation received per unit area from the sun's direct beam on a surface kept perpendicular (normal) to the sun's rays.
- `dew_point`: the temperature at which air becomes saturated with water vapor, causing condensation (dew) to form.
- `air_temperature`: the ambient temperature of the air.
- `wind_speed`: the rate at which air is moving horizontally.
- `total_precipitable_water`: the depth of water in a column of the atmosphere if all its water vapor were condensed and precipitated.
- `relative_humidity`: the percentage ratio of the current water vapor in the air to the maximum amount the air can hold at that temperature.
- `solar_zenith_angle`: the angle between the sun's rays and the vertical direction at a given location.
- `surface_albedo`: the fraction of incoming solar radiation reflected by the Earth's surface.

**Air Quality data**    The dataset was first introduced in Zhang et al. (2017) and comprises 35,064 measurements from 12 locations in Beijing, China. It includes two air-quality measurements (PM10 and PM2.5); for our experiments, we used PM10 as the target and excluded PM2.5 from the feature set. The dataset also contains ten additional environmental variables:

- $SO_2$: sulfur dioxide.
- $NO_2$: nitrogen dioxide.
- `CO`: carbon monoxide.
- $O_3$ ozone.
- `TEMP`: ambient air temperature.
- `PRES`: atmospheric pressure.
- `DEWP`: dew point temperature.
- `RAIN`: amount of precipitation.
- `wd`: the direction from which the wind is blowing. This was encoded following the method in Auer et al. (2023).
- `WSPM`: horizontal wind speed.

**Exchange rate data**    The collection of the daily exchange rates of eight foreign countries, including Australia, Britain, Canada, Switzerland, China, Japan, New Zealand, and Singapore. The data span 1990–2016, yielding 7,588 samples.

**ACEA data**    The data was first introduced in Bianchi et al. (2015). The time series represents electricity load provided by ACEA (*Azienda Comunale Energia e Ambiente*), the company that manages the electric and hydraulic distribution in the city of Rome, Italy. It was sampled every 10 minutes and contains 137,376 observations.

## B.2    POINT PREDICTORS

For each dataset, we trained three distinct point-forecasting models that span different modelling paradigms. First, a RNN built with gated recurrent unit (GRU) cells and a single layer with a hidden state size of 32. Second, a decoder-only Transformer configured with hidden size 32, a feed-forward size of 64, two attention heads, three stacked layers, and a dropout rate of 0.1. Third, an

Autoregressive Integrated Moving Average (ARIMA) model of order $(3, 1, 3)$. The first two models were trained and configured similarly to Cini et al. (2025), i.e. by minimizing the mean absolute error (MAE) loss with the Adam optimizer for 200 epochs and a batch size of 32.

For each dataset, every model was trained independently using $40\%$ of that dataset's samples as the training set. The first $25\%$ of the calibration split is used as a validation set for early stopping.

## C  HYPERPARAMETERS AND EXPERIMENTAL SETUP

### C.1  EVALUATION METRICS

Given a prediction interval $\widehat{C}^{\alpha}(\hat{y}_t) = \left[\hat{y}_t + \hat{q}_t^{\alpha/2}, \hat{y}_t + \hat{q}_t^{1-\alpha/2}\right]$ computed for a desired confidence level $\alpha$ and true observation $y_t$, we evaluated our method against the baselines using three fundamental metrics in uncertainty quantification.

**Coverage gap**   The coverage gap is defined as

$$\Delta \text{Cov} = 100 \ \left(\mathbb{1}(y_t \in \widehat{C}^{\alpha}(\hat{y}_t)) - (1 - \alpha)\right).$$

and indicates the difference between the target coverage $1 - \alpha$ and the observed one.

**Prediction interval width**   This metric quantifies the width of the prediction intervals, i.e., how uncertain the model is about prediction $\hat{y}_t$

$$\text{PI-Width} = \hat{q}_t^{1-\alpha/2} - \hat{q}_t^{\alpha/2}.$$

**Winkler score**   This is a combined metric that adds to the PI-Width a penalty if the actual value $y_t$ falls outside the PI. The penalty is proportional by a factor $\frac{2}{\alpha}$ to the degree of error of the PI, i.e. how far the actual value is from the closest bound of the interval. The Winkler score is defined as:

$$W = \begin{cases} (\hat{q}_t^{1-\alpha/2} - \hat{q}_t^{\alpha/2}) + \frac{2}{\alpha}(\hat{q}_t^{\alpha/2} - y_t) & \text{if } y_t < \hat{q}_t^{\alpha/2}, \\ (\hat{q}_t^{1-\alpha/2} - \hat{q}_t^{\alpha/2}) & \text{if } \hat{q}_t^{\alpha/2} \leq y_t \leq \hat{q}_t^{1-\alpha/2}, \\ (\hat{q}_t^{1-\alpha/2} - \hat{q}_t^{\alpha/2}) + \frac{2}{\alpha}(y_t - \hat{q}_t^{1-\alpha/2}) & \text{if } y_t > \hat{q}_t^{1-\alpha/2}. \end{cases}$$

### C.2  ESN HYPERPARAMETERS

If properly configured, ESNs can provide rich and stable representations of the temporal dependencies in the data. The main hyperparameters that we considered are: the **reservoir size**, which represents the dimension $D_h$ of the high-dimensional space where reservoir's states evolve (the "capacity" of the reservoir); the **spectral radius** $\rho(\boldsymbol{W}_h)$ of the state update weight matrix $\boldsymbol{W}_h$, which governs the stability of the internal dynamics of the reservoir; the **leak rate**, which controls the proportion of information from the previous reservoir state that is preserved and added to the new state after the non-linearity; and the **input scaling**, which regulates the magnitude of the input signal before it is projected into the reservoir, and thereby control the degree of non-linearity that will be applied to the state update.

Other hyperparameters that define how the ESN behaves are the connectivity of the reservoirs, which denotes the proportion of non-zero connections in the state update weight matrix and thereby determines the sparsity structure of the reservoir, the amount and magnitude of noise injected during state update, and the activation function for the non-linearity. Sometimes, also the topology of the reservoir can lead to meaningful dynamical properties (a ring-like arrangement of the reservoir's internal connection as opposed to random connections).

### C.3  CP-QRNN ARCHITECTURE

The architecture of CP-QRNN is identical to the model called CORNN introduced by Cini et al. (2025). It consists in a encoder-decoder architecture, in which the encoder is implemented as multiple GRU layers, and the decoder is a multilayer perceptron (MLP) that maps the learned representations to the predictions of the quantiles. The model is trained end-to-end by minimizing the pinball loss.

Table 4: Best hyperparameter values found with the grid search described in the previous section for each dataset and base model combination.

| | Model | Spectral Radius | | Leak Rate | | Input Scaling | | Temperature | Sliding Window |
|---|---|---|---|---|---|---|---|---|---|
| | | RESCP | RESCQR | RESCP | RESCQR | RESCP | RESCQR | RESCP | RESCP |
| Solar | RNN | 0.9 | 0.9 | 0.75 | 1.0 | 0.7 | 0.1 | 0.1 | 3900 |
| | Transformer | 0.9 | 0.9 | 0.8 | 1.0 | 0.4 | 0.1 | 0.1 | 7600 |
| | ARIMA | 1.0 | 0.95 | 0.8 | 0.9 | 0.2 | 0.1 | 0.1 | 1500 |
| Beijing | RNN | 1.3 | 0.99 | 0.95 | 1.0 | 0.25 | 0.1 | 0.1 | 3200 |
| | Transformer | 1.0 | 0.9 | 0.8 | 1.0 | 0.25 | 0.1 | 0.1 | 10500 |
| | ARIMA | 1.45 | 0.99 | 0.65 | 1.0 | 0.75 | 0.1 | 0.15 | 3800 |
| Exch. | RNN | 0.95 | 0.9 | 0.8 | 0.9 | 0.5 | 0.25 | 0.1 | 1000 |
| | Transformer | 0.99 | 0.99 | 0.8 | 0.9 | 0.5 | 0.1 | 0.1 | 1000 |
| | ARIMA | 0.95 | 0.99 | 0.8 | 0.8 | 0.5 | 0.1 | 0.1 | 1000 |
| ACEA | RNN | 0.95 | 0.99 | 0.8 | 0.8 | 0.25 | 0.25 | 0.1 | 1000 |
| | Transformer | 0.99 | 0.99 | 0.8 | 0.8 | 0.25 | 0.25 | 0.1 | 3000 |
| | ARIMA | 0.95 | 0.9 | 0.8 | 1.0 | 0.25 | 0.1 | 0.1 | 1000 |

## C.4 MODEL SELECTION

We performed hyperparameter tuning for all combinations of point predictors and datasets on a validation set, depending on the model.

**RESCP** For our method, we used $10\%$ of the calibration set for validation. To perform model selection, we run a grid search over the spectral radius for values in the interval $[0.5, 1.5]$, for the leak rate $[0.5, 1]$ (1 meaning no leak), for the input scaling $[0.1, 3]$, for the temperature $[0.01, 2]$ and for the size of the calibration window $[100, \text{"all"}]$, where "all" considers all states in the calibration set. The remaining ESN hyperparameters where kept fixed (reservoir size $512$, connectivity $0.2$, $\tanh$ activation function, no injected noise during state update, random connections). Given the high scalability of the method, performing such a grid search was very fast. The same was done for **RESCQR**, but only on the ESN hyperparameters (reservoir's size, spectral radius, leak rate and input scaling). In this setting, to train the readout we used the Adam optimizer (Kingma & Ba, 2015) with an initial learning rate of 0.003, which was reduced by $50\%$ every time the loss did not decrease for 10 epochs in a row. Each epoch consisted of mini-batches of 64 samples per batch.

**NexCP** The search was made for the parameter $\rho$ used to control the decay rate for the weights. The search was over the values $\{0.999, 0.99, 0.95, 0.9\}$.

**SPCI** Since SPCI (Xu & Xie, 2023a) required fitting a quantile random forest at each time step, the computational demand to use the official implementation was limiting. Therefore, we used the implementation provided by Auer et al. (2023), and followed the same protocol for the training. SPCI was run using a fixed window of 100 time steps, which corresponds to the longest window considered in Xu & Xie (2023a).

**HopCPT** We followed the same model selection procedure and hyperparameter search of the original paper (Auer et al., 2023). The model was trained for 3,000 epochs in each experiment, validating every 5 epochs on $50\%$ of the calibration data. AdamW (Loshchilov & Hutter, 2019) with standard parameters ($\beta_1 = 0.9$, $\beta_2 = 0.999$, $\delta = 0.01$) was used to optimize the model. A batch size of 4 time series was used. The model with smallest PI-Width and non-negative $\Delta$Cov was selected. If no models achieved non-negative $\Delta$Cov, then the one with the highest $\Delta$Cov was chosen.

**CP-QRNN** Also for this baseline, we followed the same model selection procedure as in the original paper (Cini et al., 2025). The number of neurons and the number of GRU layers was tuned with a small grid search on $10\%$ of the calibration data.

Table 5: Performance comparison for $\alpha = 0.05$. $\Delta$Cov values are color-coded for undercoverage cases: yellow (1-2%), orange (2-4%), red (>4%). The top three Winkler scores for each scenario are highlighted: **bold+underlined** (1st), underlined (2nd), dot-underlined (3rd).

| | | Metric | *Learning* | | | | *Non-learning* | | |
| | | | SPCI | HopCPT | CP-QRNN | **RESCQR** | SCP | NexCP | **RESCP** |
|---|---|---|---|---|---|---|---|---|---|
| Solar | RNN | $\Delta$Cov | $-0.19_{\pm0.26}$ | $-0.90_{\pm0.22}$ | $-0.33_{\pm1.48}$ | $-0.38_{\pm0.39}$ | $0.11$ | $0.97$ | $0.18_{\pm0.22}$ |
| | | PI-Width | $124.25_{\pm4.19}$ | $82.29_{\pm0.63}$ | $71.14_{\pm1.02}$ | $77.76_{\pm0.87}$ | $141.23$ | $154.71$ | $86.56_{\pm0.78}$ |
| | | Winkler | $214.33_{\pm2.03}$ | $149.88_{\pm1.15}$ | $\mathbf{\underline{94.73_{\pm0.46}}}$ | $108.42_{\pm0.34}$ | $239.24$ | $225.85$ | $138.56_{\pm1.27}$ |
| | Transf | $\Delta$Cov | $-0.40_{\pm0.26}$ | $0.50_{\pm0.64}$ | $0.48_{\pm0.78}$ | $1.05_{\pm0.12}$ | $0.06$ | $1.00$ | $-1.29_{\pm0.61}$ |
| | | PI-Width | $123.48_{\pm5.82}$ | $85.55_{\pm4.56}$ | $70.54_{\pm1.49}$ | $76.15_{\pm0.46}$ | $139.52$ | $153.24$ | $62.61_{\pm1.31}$ |
| | | Winkler | $215.15_{\pm3.01}$ | $133.84_{\pm7.86}$ | $\mathbf{\underline{93.31_{\pm0.16}}}$ | $107.31_{\pm0.35}$ | $236.43$ | $223.17$ | $105.80_{\pm2.16}$ |
| | ARIMA | $\Delta$Cov | $-0.52_{\pm0.17}$ | $0.75_{\pm1.04}$ | $-0.40_{\pm0.37}$ | $-0.62_{\pm0.15}$ | $0.07$ | $1.21$ | $0.21_{\pm0.51}$ |
| | | PI-Width | $124.61_{\pm1.37}$ | $188.97_{\pm6.52}$ | $70.67_{\pm0.55}$ | $87.34_{\pm1.33}$ | $188.57$ | $202.58$ | $99.68_{\pm1.20}$ |
| | | Winkler | $193.31_{\pm0.59}$ | $255.20_{\pm16.79}$ | $\mathbf{\underline{88.30_{\pm0.15}}}$ | $115.66_{\pm0.42}$ | $281.61$ | $265.15$ | $133.94_{\pm3.25}$ |
| Beijing | RNN | $\Delta$Cov | $-1.38_{\pm1.13}$ | $-1.06_{\pm1.40}$ | $-0.36_{\pm0.55}$ | $-0.58_{\pm0.38}$ | $-0.18$ | $-0.01$ | $-0.60_{\pm0.17}$ |
| | | PI-Width | $97.22_{\pm6.98}$ | $112.10_{\pm31.41}$ | $84.51_{\pm2.90}$ | $87.50_{\pm3.98}$ | $98.51$ | $100.48$ | $86.14_{\pm0.95}$ |
| | | Winkler | $167.81_{\pm2.37}$ | $183.15_{\pm20.62}$ | $130.62_{\pm0.69}$ | $136.65_{\pm1.07}$ | $170.36$ | $166.04$ | $\mathbf{\underline{124.24_{\pm0.11}}}$ |
| | Transf | $\Delta$Cov | $-1.27_{\pm0.46}$ | $-0.46_{\pm0.56}$ | $-0.43_{\pm0.63}$ | $-1.05_{\pm0.83}$ | $-0.27$ | $-0.03$ | $-0.52_{\pm0.32}$ |
| | | PI-Width | $95.82_{\pm5.28}$ | $100.94_{\pm6.20}$ | $84.24_{\pm3.84}$ | $87.66_{\pm6.01}$ | $97.75$ | $100.17$ | $82.11_{\pm1.88}$ |
| | | Winkler | $167.34_{\pm2.11}$ | $167.90_{\pm2.74}$ | $131.27_{\pm1.09}$ | $137.80_{\pm2.26}$ | $171.09$ | $166.74$ | $\mathbf{\underline{119.46_{\pm0.35}}}$ |
| | ARIMA | $\Delta$Cov | $-0.53_{\pm0.36}$ | $-1.20_{\pm0.45}$ | $-1.30_{\pm0.36}$ | $-1.13_{\pm0.49}$ | $-0.23$ | $-0.14$ | $-0.29_{\pm0.22}$ |
| | | PI-Width | $104.81_{\pm3.69}$ | $94.47_{\pm2.13}$ | $81.64_{\pm3.94}$ | $85.17_{\pm1.77}$ | $106.81$ | $107.73$ | $94.54_{\pm1.67}$ |
| | | Winkler | $175.14_{\pm2.04}$ | $168.88_{\pm1.21}$ | $\mathbf{\underline{128.63_{\pm1.34}}}$ | $136.63_{\pm1.26}$ | $180.18$ | $174.82$ | $154.79_{\pm0.33}$ |
| Exchange | RNN | $\Delta$Cov | $0.58_{\pm0.31}$ | $1.72_{\pm0.00}$ | $-2.00_{\pm2.22}$ | $-0.52_{\pm3.06}$ | $1.35$ | $1.18$ | $0.54_{\pm0.20}$ |
| | | PI-Width | $0.0290_{\pm0.0008}$ | $0.0513_{\pm0.0000}$ | $0.0404_{\pm0.0011}$ | $0.0479_{\pm0.0019}$ | $0.0549$ | $0.0497$ | $0.0260_{\pm0.0002}$ |
| | | Winkler | $0.0356_{\pm0.0005}$ | $0.0608_{\pm0.0000}$ | $0.0568_{\pm0.0002}$ | $0.0578_{\pm0.0009}$ | $0.0645$ | $0.0604$ | $\mathbf{\underline{0.0321_{\pm0.0004}}}$ |
| | Transf | $\Delta$Cov | $1.98_{\pm0.38}$ | $1.81_{\pm0.00}$ | $0.94_{\pm1.00}$ | $-0.01_{\pm2.96}$ | $2.67$ | $1.93$ | $0.67_{\pm0.11}$ |
| | | PI-Width | $0.0310_{\pm0.0008}$ | $0.0501_{\pm0.0000}$ | $0.0441_{\pm0.0015}$ | $0.0463_{\pm0.0044}$ | $0.0646$ | $0.0601$ | $0.0281_{\pm0.0001}$ |
| | | Winkler | $0.0375_{\pm0.0006}$ | $0.0600_{\pm0.0000}$ | $0.0585_{\pm0.0006}$ | $0.0595_{\pm0.0011}$ | $0.0745$ | $0.0712$ | $\mathbf{\underline{0.0355_{\pm0.0002}}}$ |
| | ARIMA | $\Delta$Cov | $1.01_{\pm0.45}$ | $1.39_{\pm0.00}$ | $-1.50_{\pm0.75}$ | $-1.76_{\pm0.63}$ | $1.87$ | $1.32$ | $-0.03_{\pm0.15}$ |
| | | PI-Width | $0.0293_{\pm0.0005}$ | $0.0482_{\pm0.0000}$ | $0.0404_{\pm0.0006}$ | $0.0419_{\pm0.0016}$ | $0.0489$ | $0.0442$ | $0.0258_{\pm0.0000}$ |
| | | Winkler | $0.0357_{\pm0.0004}$ | $0.0576_{\pm0.0000}$ | $0.0560_{\pm0.0004}$ | $0.0547_{\pm0.0007}$ | $0.0585$ | $0.0552$ | $\mathbf{\underline{0.0328_{\pm0.0001}}}$ |
| ACEA | RNN | $\Delta$Cov | $-1.20_{\pm0.96}$ | $-1.92_{\pm0.00}$ | $-8.29_{\pm5.64}$ | $-15.32_{\pm4.57}$ | $-0.89$ | $-0.29$ | $0.32_{\pm0.22}$ |
| | | PI-Width | $11.13_{\pm0.88}$ | $23.20_{\pm0.00}$ | $19.85_{\pm2.59}$ | $18.18_{\pm1.58}$ | $24.31$ | $24.48$ | $11.90_{\pm0.30}$ |
| | | Winkler | $18.38_{\pm0.40}$ | $32.48_{\pm0.00}$ | $39.72_{\pm4.66}$ | $46.11_{\pm6.34}$ | $32.57$ | $31.23$ | $\mathbf{\underline{16.31_{\pm0.23}}}$ |
| | Transf | $\Delta$Cov | $-1.75_{\pm1.19}$ | $-2.04_{\pm0.00}$ | $-10.72_{\pm7.28}$ | $-21.13_{\pm7.72}$ | $-3.49$ | $-0.27$ | $1.72_{\pm0.06}$ |
| | | PI-Width | $11.22_{\pm0.48}$ | $22.80_{\pm0.00}$ | $18.44_{\pm2.29}$ | $16.25_{\pm2.03}$ | $21.99$ | $25.34$ | $12.62_{\pm0.37}$ |
| | | Winkler | $18.84_{\pm0.67}$ | $32.54_{\pm0.00}$ | $41.67_{\pm9.46}$ | $52.93_{\pm8.53}$ | $34.43$ | $32.01$ | $\mathbf{\underline{16.19_{\pm0.40}}}$ |
| | ARIMA | $\Delta$Cov | $0.92_{\pm0.37}$ | $-1.87_{\pm0.00}$ | $-17.36_{\pm11.98}$ | $-23.84_{\pm7.89}$ | $-0.13$ | $-0.24$ | $-2.70_{\pm0.76}$ |
| | | PI-Width | $14.86_{\pm0.40}$ | $39.74_{\pm0.00}$ | $25.26_{\pm4.44}$ | $20.67_{\pm2.38}$ | $43.65$ | $41.42$ | $11.68_{\pm0.76}$ |
| | | Winkler | $21.83_{\pm0.19}$ | $49.57_{\pm0.00}$ | $63.53_{\pm17.63}$ | $67.04_{\pm12.20}$ | $50.52$ | $48.69$ | $\mathbf{\underline{18.01_{\pm0.53}}}$ |

## C.5 Resulting hyperparameters

The best hyperparameters found with the grid search described in the previous section can be found in Tab. 4.

## D Additional results

### D.1 Results with different significance levels

In addition to the results presented in the main text for miscoverage level $\alpha = 0.1$, we also evaluate the performance of RESCP and RESCQR at significance levels $\alpha = 0.05$ and $\alpha = 0.15$. Tables 5 and 6 summarize the results for these additional significance levels across all datasets and base forecasters and against all the baselines considered in the main text. The results are consistent with those discussed in the main text for $\alpha = 0.1$, confirming the effectiveness of RESCP in producing well-

Table 6: Performance comparison for $\alpha = 0.15$. $\Delta$Cov values are color-coded for undercoverage cases: yellow (1-2%), orange (2-4%), red (>4%). The top three Winkler scores for each scenario are highlighted: **bold+underlined** (1st), underlined (2nd), dot-underlined (3rd).

| | | Metric | Learning | | | | Non-learning | | |
| --- | --- | --- | --- | --- | --- | --- | --- | --- | --- |
| | | | SPCI | HopCPT | CP-QRNN | **RESCQR** | SCP | NexCP | **RESCP** |
| Solar | RNN | $\Delta$Cov | $0.72_{\pm0.23}$ | $-1.77_{\pm0.33}$ | $-1.10_{\pm1.83}$ | $0.10_{\pm1.03}$ | 0.64 | 1.82 | $-0.47_{\pm0.32}$ |
| | | PI-Width | $46.78_{\pm0.81}$ | $47.68_{\pm0.46}$ | $46.84_{\pm0.46}$ | $47.43_{\pm0.56}$ | 47.85 | 55.73 | $46.37_{\pm0.36}$ |
| | | Winkler | $119.10_{\pm0.51}$ | $96.70_{\pm0.32}$ | $\mathbf{69.15_{\pm0.14}}$ | $75.01_{\pm0.11}$ | 134.09 | 130.47 | $87.44_{\pm0.22}$ |
| | Transf | $\Delta$Cov | $0.34_{\pm0.31}$ | $2.88_{\pm0.72}$ | $2.54_{\pm4.23}$ | $-3.19_{\pm16.62}$ | 0.60 | 1.81 | $0.18_{\pm0.66}$ |
| | | PI-Width | $46.56_{\pm1.69}$ | $46.78_{\pm2.76}$ | $46.47_{\pm1.25}$ | $47.06_{\pm0.36}$ | 48.72 | 56.72 | $39.37_{\pm0.51}$ |
| | | Winkler | $119.81_{\pm0.62}$ | $89.44_{\pm3.30}$ | $\mathbf{68.48_{\pm0.14}}$ | $74.44_{\pm0.21}$ | 133.18 | 129.62 | $75.69_{\pm0.39}$ |
| | ARIMA | $\Delta$Cov | $0.87_{\pm0.43}$ | $1.98_{\pm2.56}$ | $-1.06_{\pm0.43}$ | $-0.43_{\pm0.40}$ | -0.07 | 2.29 | $0.30_{\pm0.61}$ |
| | | PI-Width | $71.25_{\pm1.42}$ | $111.82_{\pm7.83}$ | $50.18_{\pm0.44}$ | $60.41_{\pm1.26}$ | 93.98 | 102.74 | $62.24_{\pm0.94}$ |
| | | Winkler | $124.75_{\pm0.35}$ | $171.47_{\pm11.32}$ | $\mathbf{68.52_{\pm0.09}}$ | $87.26_{\pm0.24}$ | 179.83 | 174.48 | $98.54_{\pm4.21}$ |
| Beijing | RNN | $\Delta$Cov | $-0.94_{\pm1.14}$ | $0.56_{\pm4.00}$ | $-1.31_{\pm1.19}$ | $-0.26_{\pm1.93}$ | -0.48 | 0.10 | $-0.98_{\pm1.17}$ |
| | | PI-Width | $55.46_{\pm2.03}$ | $67.60_{\pm18.87}$ | $51.21_{\pm1.80}$ | $54.58_{\pm3.94}$ | 52.93 | 55.07 | $52.21_{\pm2.06}$ |
| | | Winkler | $103.43_{\pm0.64}$ | $112.44_{\pm11.23}$ | $87.49_{\pm0.38}$ | $90.42_{\pm0.38}$ | 104.56 | 103.16 | $\mathbf{86.72_{\pm0.12}}$ |
| | Transf | $\Delta$Cov | $-0.69_{\pm0.57}$ | $-0.45_{\pm1.23}$ | $-1.32_{\pm1.71}$ | $-1.68_{\pm1.81}$ | -0.46 | 0.11 | $-0.75_{\pm1.35}$ |
| | | PI-Width | $54.51_{\pm1.07}$ | $55.14_{\pm3.17}$ | $50.67_{\pm2.82}$ | $52.98_{\pm3.10}$ | 52.42 | 54.64 | $50.31_{\pm2.16}$ |
| | | Winkler | $102.74_{\pm0.45}$ | $103.53_{\pm0.85}$ | $87.71_{\pm0.41}$ | $90.97_{\pm1.12}$ | 104.60 | 103.21 | $\mathbf{84.47_{\pm0.23}}$ |
| | ARIMA | $\Delta$Cov | $0.74_{\pm0.30}$ | $-3.08_{\pm0.97}$ | $-2.04_{\pm0.89}$ | $-2.07_{\pm0.79}$ | -0.25 | -0.09 | $-0.32_{\pm0.46}$ |
| | | PI-Width | $60.45_{\pm0.85}$ | $53.46_{\pm1.10}$ | $50.58_{\pm1.85}$ | $51.64_{\pm1.31}$ | 60.29 | 60.73 | $54.70_{\pm1.13}$ |
| | | Winkler | $108.16_{\pm0.47}$ | $107.50_{\pm0.68}$ | $\mathbf{86.46_{\pm0.55}}$ | $90.85_{\pm0.38}$ | 112.79 | 110.66 | $99.59_{\pm0.30}$ |
| Exchange | RNN | $\Delta$Cov | $5.13_{\pm0.59}$ | $3.05_{\pm0.00}$ | $-3.28_{\pm2.93}$ | $2.40_{\pm1.73}$ | 3.06 | 2.26 | $1.77_{\pm0.29}$ |
| | | PI-Width | $0.0207_{\pm0.0004}$ | $0.0342_{\pm0.0000}$ | $0.0284_{\pm0.0009}$ | $0.0323_{\pm0.0010}$ | 0.0380 | 0.0351 | $0.0180_{\pm0.0001}$ |
| | | Winkler | $0.0247_{\pm0.0002}$ | $0.0420_{\pm0.0000}$ | $0.0413_{\pm0.0005}$ | $0.0410_{\pm0.0006}$ | 0.0452 | 0.0436 | $\mathbf{0.0234_{\pm0.0002}}$ |
| | Transf | $\Delta$Cov | $6.43_{\pm0.46}$ | $3.78_{\pm0.00}$ | $0.25_{\pm1.86}$ | $-2.20_{\pm5.75}$ | 6.03 | 4.08 | $2.37_{\pm0.15}$ |
| | | PI-Width | $0.0221_{\pm0.0005}$ | $0.0341_{\pm0.0000}$ | $0.0301_{\pm0.0007}$ | $0.0313_{\pm0.0016}$ | 0.0483 | 0.0455 | $0.0195_{\pm0.0000}$ |
| | | Winkler | $\mathbf{0.0261_{\pm0.0004}}$ | $0.0421_{\pm0.0000}$ | $0.0425_{\pm0.0006}$ | $0.0422_{\pm0.0003}$ | 0.0558 | 0.0545 | $0.0262_{\pm0.0001}$ |
| | ARIMA | $\Delta$Cov | $6.23_{\pm0.72}$ | $2.96_{\pm0.00}$ | $-2.26_{\pm1.49}$ | $-2.04_{\pm3.06}$ | 4.25 | 2.77 | $1.03_{\pm0.36}$ |
| | | PI-Width | $0.0213_{\pm0.0009}$ | $0.0322_{\pm0.0000}$ | $0.0279_{\pm0.0006}$ | $0.0294_{\pm0.0016}$ | 0.0330 | 0.0303 | $0.0175_{\pm0.0001}$ |
| | | Winkler | $0.0253_{\pm0.0006}$ | $0.0399_{\pm0.0000}$ | $0.0405_{\pm0.0005}$ | $0.0401_{\pm0.0003}$ | 0.0404 | $0.0392$ | $\mathbf{0.0234_{\pm0.0001}}$ |
| ACEA | RNN | $\Delta$Cov | $-1.84_{\pm4.08}$ | $-2.43_{\pm0.00}$ | $-16.15_{\pm9.30}$ | $-20.50_{\pm5.38}$ | -1.55 | -0.46 | $1.86_{\pm0.40}$ |
| | | PI-Width | $7.42_{\pm0.71}$ | $15.93_{\pm0.00}$ | $13.09_{\pm1.44}$ | $13.29_{\pm1.31}$ | 16.34 | 17.07 | $8.06_{\pm0.13}$ |
| | | Winkler | $12.33_{\pm0.15}$ | $24.51_{\pm0.00}$ | $28.71_{\pm3.73}$ | $30.47_{\pm2.40}$ | 24.66 | $24.15$ | $\mathbf{11.14_{\pm0.11}}$ |
| | Transf | $\Delta$Cov | $-2.58_{\pm0.78}$ | $-2.98_{\pm0.00}$ | $-17.58_{\pm10.86}$ | $-29.97_{\pm8.62}$ | -5.76 | -0.56 | $4.20_{\pm0.23}$ |
| | | PI-Width | $7.34_{\pm0.12}$ | $15.26_{\pm0.00}$ | $11.98_{\pm1.39}$ | $10.99_{\pm1.51}$ | 13.66 | 16.73 | $8.58_{\pm0.11}$ |
| | | Winkler | $12.69_{\pm0.10}$ | $24.40_{\pm0.00}$ | $29.59_{\pm5.71}$ | $33.91_{\pm3.65}$ | 25.63 | 24.52 | $\mathbf{11.42_{\pm0.20}}$ |
| | ARIMA | $\Delta$Cov | $0.39_{\pm1.05}$ | $-4.81_{\pm0.00}$ | $-36.21_{\pm8.89}$ | $-34.84_{\pm9.41}$ | -1.27 | -0.72 | $1.16_{\pm1.10}$ |
| | | PI-Width | $10.92_{\pm0.29}$ | $30.89_{\pm0.00}$ | $14.51_{\pm1.64}$ | $13.39_{\pm1.91}$ | 33.68 | 32.72 | $8.31_{\pm0.52}$ |
| | | Winkler | $15.51_{\pm0.12}$ | $41.82_{\pm0.00}$ | $47.54_{\pm5.69}$ | $44.95_{\pm6.32}$ | 43.00 | $40.68$ | $\mathbf{11.66_{\pm0.51}}$ |

calibrated and accurate PIs across different significance levels, datasets, and base point predictors.

## D.2 CALIBRATION CURVES

To better evaluate the calibration performance of RESCP and RESCQR, we plot their calibration curves for all datasets and base models. The results are shown in Fig. 2. The calibration curve plots the target coverage against the observed coverage, where a perfectly calibrated method would follow the diagonal line $y = x$, which is depicted as a black dashed line in the plots. The target coverages $\{0.7, 0.75, 0.8, 0.85, 0.9, 0.95\}$, are shown on the $x$-axis, and the corresponding observed coverages are shown on the $y$-axis. Each row of the grid corresponds to a dataset, while each column to a base point forecaster. The results confirm the findings discussed in the main text for $\alpha = 0.1$, showing that RESCP produces well-calibrated PIs across different significance levels, datasets, and base models. RESCQR also generally generates accurate prediction intervals, but its performance are poor in some benchmarks such as ACEA.

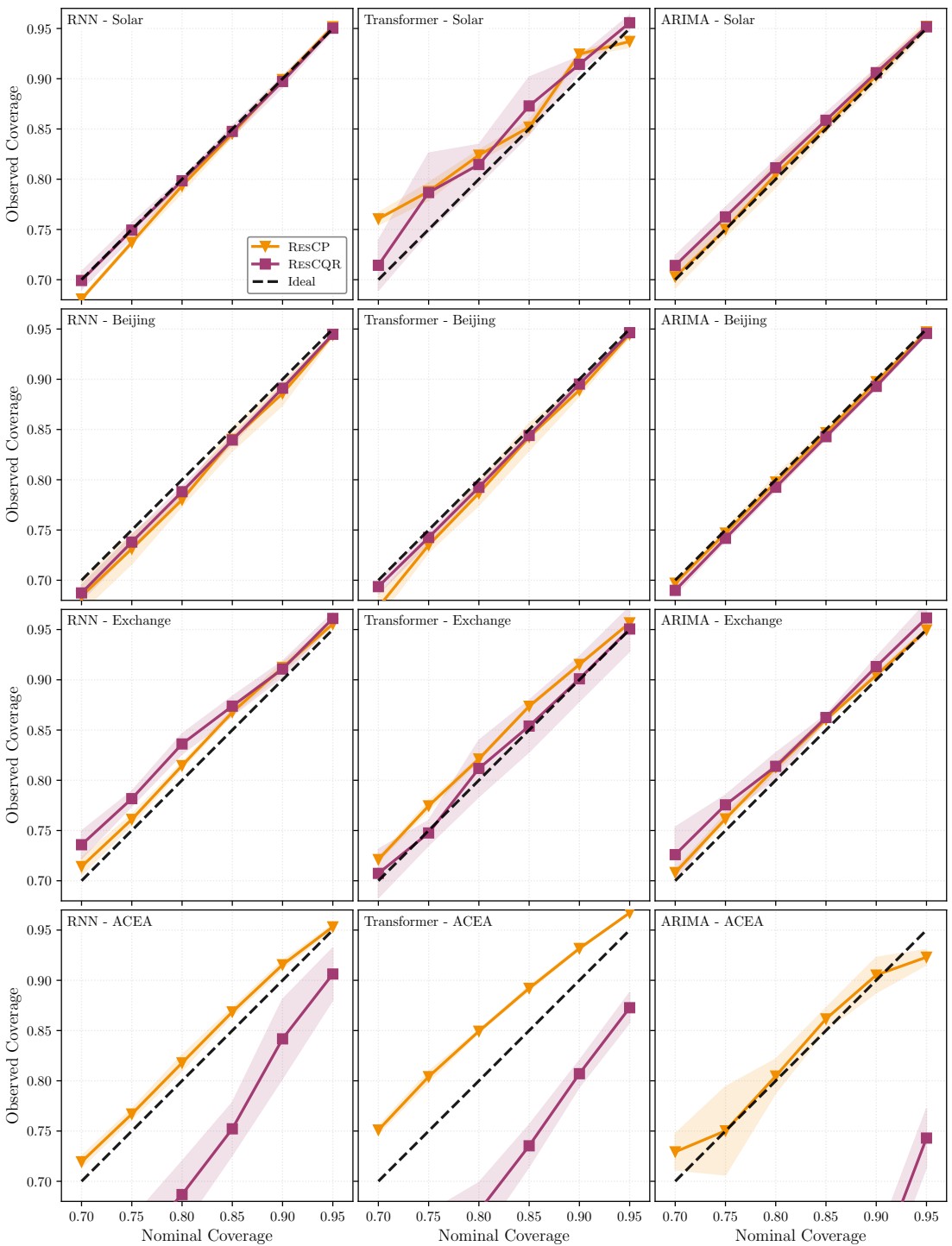

Figure 3: Calibration curves of RESCP and RESCQR over all datasets and baselines.

Table 7: Ablation study on the AR synthetic dataset for three miscoverage levels ($\alpha \in \{0.05, 0.1, 0.15\}$).

| $\alpha$ | Metric | **RESCP** | No decay | No window | No decay, no window | NExCP | SCP |
|---|---|---|---|---|---|---|---|
| 0.05 | Coverage | $94.41_{\pm0.21}$ | $93.50_{\pm0.23}$ | $91.62_{\pm0.27}$ | $88.39_{\pm0.17}$ | 94.34 | 86.11 |
| 0.05 | $\Delta$Cov | $-0.59_{\pm0.21}$ | $-1.50_{\pm0.23}$ | $-3.38_{\pm0.27}$ | $-6.61_{\pm0.17}$ | -0.66 | -8.89 |
| 0.1 | Coverage | $89.75_{\pm0.28}$ | $88.25_{\pm0.41}$ | $84.45_{\pm0.19}$ | $80.85_{\pm0.17}$ | 88.84 | 78.48 |
| 0.1 | $\Delta$Cov | $-0.25_{\pm0.28}$ | $-1.75_{\pm0.41}$ | $-5.55_{\pm0.19}$ | $-9.15_{\pm0.17}$ | -1.16 | -11.52 |
| 0.15 | Coverage | $84.93_{\pm0.34}$ | $83.11_{\pm0.35}$ | $78.33_{\pm0.20}$ | $73.99_{\pm0.28}$ | 84.14 | 71.92 |
| 0.15 | $\Delta$Cov | $-0.07_{\pm0.34}$ | $-1.89_{\pm0.35}$ | $-6.67_{\pm0.20}$ | $-11.01_{\pm0.28}$ | -0.86 | -13.08 |

## D.3 SYNTHETIC DATASET

To further validate the effectiveness of RESCP, we conduct an ablation study on synthetic data where the ground truth is known. We generate 10000 samples from a synthetic autoregressive (AR) process of order 1

$$y_t = \phi_t y_{t-1} + \epsilon_t \quad \text{with} \quad \epsilon_t \sim \mathcal{N}(0,1)$$

with time-varying coefficients $\phi_t$ across the calibration and test sets. Specifically, we introduce three change points in the coefficient sequence $\phi$ between the calibration and test sets. Overall, $\phi_t$ assumes the values $\{-0.9, 0.3, -0.5, 0.7\}$. The first change point occurs at the midpoint of the calibration set, while the remaining two are placed at equal intervals over the rest of the series. The same setup as in Sec. 4 is used: data is split into training, calibration, and test sets with proportions $40\%/40\%/20\%$, and an RNN is employed as the base forecaster. We then compare the performance of RESCP by removing its two key components: (i) the time-dependent weights, (ii) the sliding window of past residuals used for calibration, and (iii) the combination of the two. The results, reported in Tab. 7, demonstrate that both components significantly contribute to the performance of RESCP, with the full model achieving the best results in terms of coverage. This ablation study on synthetic data further confirms the robustness and effectiveness of RESCP in capturing the underlying dynamics of time series data.

## D.4 VISUALIZATION OF THE ADAPTIVE PIS

To give a better intuition of how RESCP adapts the width of the PIs over time in response to changes in the data distribution, we visualize some examples of adaptive PIs produced by RESCP on the ACEA and Solar datasets in Fig. 4 and Fig. 5, respectively. These plots illustrate how RESCP effectively adjusts the uncertainty quantification based on the observed patterns.

## E RESCP ABLATION STUDY

### E.1 ABLATION ON RESCP COMPONENTS

In this section, we report the remaining results of the ablation study on RESCP components presented in Sec. 4 for the RNN base model, extending them to all base models considered in our experiments. The results, summarized in Tab. 8, confirm the findings discussed in the main text. Specifically, we see that both the time-dependent weights and the sliding window of past residuals significantly contribute to the performance of RESCP across all datasets and base models.

### E.2 USE OF EXOGENOUS VARIABLES

We further investigate the impact of incorporating exogenous variables into the RESCP framework for the Solar and Beijing datasets. To this end, we conduct experiments comparing the performance of RESCP with and without exogenous inputs using all base models. The results, summarized in Tab. 9, show that the inclusion of exogenous variables in RESCP generally leads to worse performance in terms of coverage and Winkler score. This supports our choice of excluding exogenous variables from RESCP. Removing exogenous variables from RESCQR, instead, leads to clear degraded performances on the Solar dataset, while on the Beijing dataset the results are similar (exogenous inputs do not appear particularly relevant to quantify uncertainty in this dataset).

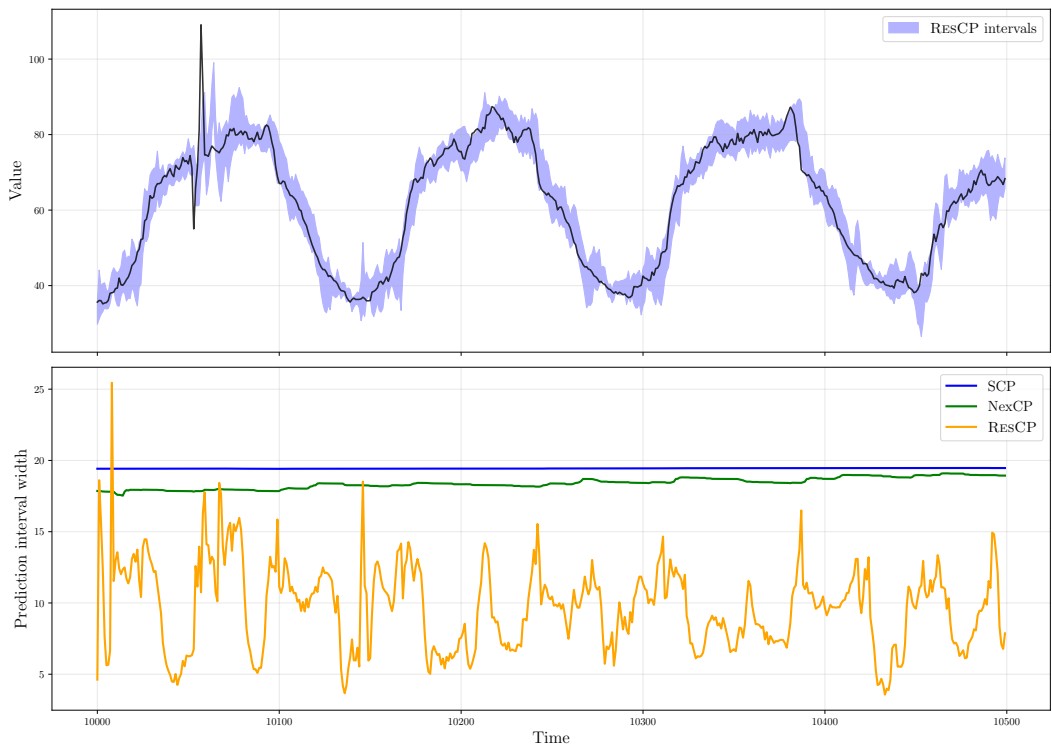

Figure 4: Prediction intervals of RESCP (top) and prediction intervals' width of different methods (bottom) for the ACEA dataset

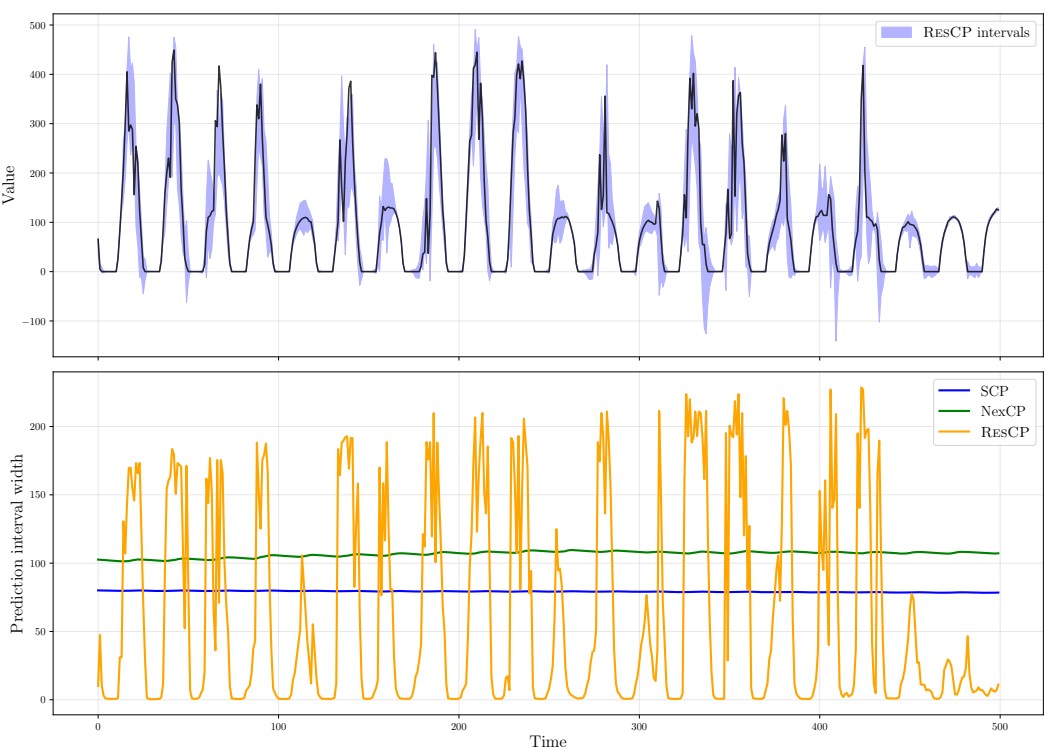

Figure 5: Prediction intervals of RESCP (top) and prediction intervals' width of different methods (bottom) for the Solar dataset

Table 8: Ablation of **RESCP**, RESCP without decay, RESCP without sliding window, and the combination of the two. Best Winkler scores are shown in bold.

| | | Metric | **RESCP** | No decay | No window | No window, no decay |
|---|---|---|---|---|---|---|
| Solar | RNN | $\Delta$Cov | $0.74_{\pm 0.24}$ | $-0.10_{\pm 0.26}$ | $0.89_{\pm 0.20}$ | $1.70_{\pm 0.10}$ |
| | | PI-Width | $62.25_{\pm 0.75}$ | $60.59_{\pm 0.84}$ | $59.81_{\pm 0.87}$ | $60.90_{\pm 0.30}$ |
| | | Winkler | $\mathbf{104.24_{\pm 0.79}}$ | $107.46_{\pm 0.49}$ | $104.70_{\pm 0.20}$ | $104.46_{\pm 0.40}$ |
| | Transf | $\Delta$Cov | $3.09_{\pm 0.35}$ | $2.22_{\pm 0.37}$ | $-0.37_{\pm 0.42}$ | $0.73_{\pm 0.54}$ |
| | | PI-Width | $63.34_{\pm 1.11}$ | $58.49_{\pm 0.82}$ | $53.41_{\pm 0.74}$ | $56.37_{\pm 0.69}$ |
| | | Winkler | $103.13_{\pm 0.58}$ | $103.35_{\pm 0.94}$ | $103.71_{\pm 0.39}$ | $\mathbf{101.36_{\pm 0.43}}$ |
| | ARIMA | $\Delta$Cov | $0.68_{\pm 0.95}$ | $-1.06_{\pm 1.27}$ | $2.93_{\pm 0.34}$ | $2.97_{\pm 0.78}$ |
| | | PI-Width | $77.17_{\pm 2.07}$ | $80.27_{\pm 2.90}$ | $82.68_{\pm 2.29}$ | $85.91_{\pm 2.82}$ |
| | | Winkler | $\mathbf{110.38_{\pm 4.03}}$ | $113.09_{\pm 7.35}$ | $114.41_{\pm 1.98}$ | $118.20_{\pm 2.85}$ |
| Beijing | RNN | $\Delta$Cov | $-0.70_{\pm 0.77}$ | $-1.64_{\pm 0.61}$ | $0.10_{\pm 1.36}$ | $1.64_{\pm 0.88}$ |
| | | PI-Width | $65.96_{\pm 2.50}$ | $64.04_{\pm 1.69}$ | $64.91_{\pm 3.36}$ | $69.53_{\pm 3.02}$ |
| | | Winkler | $106.07_{\pm 0.47}$ | $114.85_{\pm 0.29}$ | $\mathbf{97.99_{\pm 0.31}}$ | $98.25_{\pm 0.66}$ |
| | Transf | $\Delta$Cov | $-0.49_{\pm 0.59}$ | $-1.53_{\pm 0.75}$ | $-1.45_{\pm 0.73}$ | $-1.09_{\pm 0.70}$ |
| | | PI-Width | $64.06_{\pm 1.74}$ | $62.23_{\pm 2.17}$ | $62.37_{\pm 1.96}$ | $63.94_{\pm 2.01}$ |
| | | Winkler | $\mathbf{103.64_{\pm 0.21}}$ | $112.31_{\pm 0.74}$ | $111.88_{\pm 0.50}$ | $112.40_{\pm 0.51}$ |
| | ARIMA | $\Delta$Cov | $0.63_{\pm 0.22}$ | $0.35_{\pm 0.23}$ | $1.31_{\pm 0.62}$ | $2.36_{\pm 0.41}$ |
| | | PI-Width | $70.43_{\pm 0.86}$ | $70.90_{\pm 0.51}$ | $72.85_{\pm 2.17}$ | $77.26_{\pm 1.67}$ |
| | | Winkler | $\mathbf{108.75_{\pm 0.31}}$ | $119.72_{\pm 0.44}$ | $118.14_{\pm 0.69}$ | $118.50_{\pm 0.46}$ |
| Exch. | RNN | $\Delta$Cov | $1.13_{\pm 0.27}$ | $2.01_{\pm 0.19}$ | $4.19_{\pm 0.05}$ | $4.20_{\pm 0.12}$ |
| | | PI-Width | $0.0210_{\pm 0.0001}$ | $0.0219_{\pm 0.0001}$ | $0.0249_{\pm 0.0002}$ | $0.0254_{\pm 0.0003}$ |
| | | Winkler | $\mathbf{0.0264_{\pm 0.0002}}$ | $0.0269_{\pm 0.0002}$ | $0.0284_{\pm 0.0001}$ | $0.0291_{\pm 0.0002}$ |
| | Transf | $\Delta$Cov | $1.46_{\pm 0.18}$ | $2.36_{\pm 0.07}$ | $4.45_{\pm 0.08}$ | $4.80_{\pm 0.14}$ |
| | | PI-Width | $0.0229_{\pm 0.0001}$ | $0.0241_{\pm 0.0002}$ | $0.0265_{\pm 0.0003}$ | $0.0267_{\pm 0.0002}$ |
| | | Winkler | $\mathbf{0.0294_{\pm 0.0001}}$ | $0.0298_{\pm 0.0002}$ | $0.0306_{\pm 0.0003}$ | $0.0307_{\pm 0.0003}$ |
| | ARIMA | $\Delta$Cov | $0.38_{\pm 0.41}$ | $1.61_{\pm 0.21}$ | $3.90_{\pm 0.21}$ | $3.40_{\pm 0.39}$ |
| | | PI-Width | $0.0207_{\pm 0.0001}$ | $0.0215_{\pm 0.0001}$ | $0.0245_{\pm 0.0001}$ | $0.0248_{\pm 0.0002}$ |
| | | Winkler | $\mathbf{0.0268_{\pm 0.0001}}$ | $0.0273_{\pm 0.0003}$ | $0.0288_{\pm 0.0002}$ | $0.0293_{\pm 0.0002}$ |
| ACEA | RNN | $\Delta$Cov | $1.56_{\pm 0.62}$ | $2.79_{\pm 0.45}$ | $5.34_{\pm 0.38}$ | $4.96_{\pm 0.63}$ |
| | | PI-Width | $9.61_{\pm 0.26}$ | $10.15_{\pm 0.09}$ | $11.88_{\pm 0.09}$ | $12.15_{\pm 0.17}$ |
| | | Winkler | $\mathbf{12.91_{\pm 0.23}}$ | $13.41_{\pm 0.08}$ | $14.80_{\pm 0.17}$ | $15.25_{\pm 0.40}$ |
| | Transf | $\Delta$Cov | $3.54_{\pm 0.32}$ | $3.29_{\pm 0.35}$ | $2.97_{\pm 0.47}$ | $3.04_{\pm 0.21}$ |
| | | PI-Width | $10.10_{\pm 0.16}$ | $10.32_{\pm 0.05}$ | $9.34_{\pm 0.15}$ | $9.45_{\pm 0.07}$ |
| | | Winkler | $12.90_{\pm 0.16}$ | $13.45_{\pm 0.15}$ | $\mathbf{12.27_{\pm 0.26}}$ | $12.49_{\pm 0.10}$ |
| | ARIMA | $\Delta$Cov | $5.02_{\pm 0.40}$ | $5.17_{\pm 0.33}$ | $7.68_{\pm 0.22}$ | $7.75_{\pm 0.24}$ |
| | | PI-Width | $13.63_{\pm 0.55}$ | $13.50_{\pm 0.32}$ | $16.74_{\pm 0.72}$ | $16.91_{\pm 0.50}$ |
| | | Winkler | $\mathbf{16.21_{\pm 0.53}}$ | $16.27_{\pm 0.45}$ | $19.24_{\pm 0.64}$ | $19.44_{\pm 0.45}$ |

# F  SENSITIVITY ANALYSIS ON ESN HYPERPARAMETERS AND TEMPERATURE

After performing model selection and evaluating our model with the resulting hyperparameters, we conducted a small sensitivity analysis to determine how each part of the reservoir influenced model performance in RESCP, allowing us to assess the robustness of our chosen hyperparameters and identify the key aspects of the reservoir architecture that contribute most to quantification accuracy. We considered the RNN point forecasting baseline.

Fig. 6 shows how the value of the Winkler score and of the coverage varies as the temperature used in the SOFTMAX in equation Eq. 7 increases. As can be seen, there is a well-defined optimal range for the temperature in which the Winkler score is minimal and the coverage is close to the targeted one. Notice that for a small temperature, the method is unable to achieve the nominal coverage. This is the *bias-variace tradeoff* that we mentioned in Section 3.1.1, more specifically in Assumption 3.5: the temperature should be small enough to localize only the meaningful residuals close to the query one, yet high enough to avoid collapsing the neighborhood to a single or limited set of similar samples.

Fig. 7 shows the behaviour of RESCP depending on the capacity of the reservoir. Apart from the Beijing dataset, the method can achieve the nominal coverage also when the dimensionality of the

Table 9: Comparison of **RESCP** and **RESCQR** with and without exogenous variables for Solar and Beijing datasets. Best Winkler scores (lowest) within each method are shown in bold.

|  |  | Metric | **RESCP** | w/ exog | **RESCQR** | w/o exog |
|---|---|---|---|---|---|---|
| Solar | RNN | $\Delta$Cov | $0.74_{\pm 0.24}$ | $1.34_{\pm 0.09}$ | $-1.10_{\pm 0.91}$ | $-0.70_{\pm 0.12}$ |
|  |  | PI-Width | $62.25_{\pm 0.75}$ | $53.87_{\pm 0.59}$ | $59.99_{\pm 1.72}$ | $56.71_{\pm 0.17}$ |
|  |  | Winkler | $\mathbf{104.24_{\pm 0.79}}$ | $105.55_{\pm 0.81}$ | $\mathbf{82.76_{\pm 0.26}}$ | $88.46_{\pm 0.42}$ |
|  | Transf | $\Delta$Cov | $3.09_{\pm 0.35}$ | $3.57_{\pm 0.02}$ | $-3.51_{\pm 16.26}$ | $1.37_{\pm 1.08}$ |
|  |  | PI-Width | $63.34_{\pm 1.11}$ | $56.96_{\pm 0.27}$ | $59.56_{\pm 1.59}$ | $57.84_{\pm 0.36}$ |
|  |  | Winkler | $103.13_{\pm 0.58}$ | $\mathbf{101.73_{\pm 0.33}}$ | $\mathbf{82.16_{\pm 0.32}}$ | $88.30_{\pm 0.53}$ |
|  | ARIMA | $\Delta$Cov | $0.68_{\pm 0.95}$ | $-1.12_{\pm 0.13}$ | $-2.03_{\pm 0.62}$ | $-1.03_{\pm 0.59}$ |
|  |  | PI-Width | $77.17_{\pm 2.07}$ | $78.09_{\pm 0.47}$ | $66.19_{\pm 0.81}$ | $66.85_{\pm 0.93}$ |
|  |  | Winkler | $\mathbf{110.38_{\pm 4.03}}$ | $127.79_{\pm 1.00}$ | $\mathbf{85.38_{\pm 0.45}}$ | $88.19_{\pm 0.89}$ |
| Beijing | RNN | $\Delta$Cov | $-0.70_{\pm 0.77}$ | $-4.71_{\pm 0.14}$ | $-1.21_{\pm 1.65}$ | $-0.53_{\pm 0.26}$ |
|  |  | PI-Width | $65.96_{\pm 2.50}$ | $56.29_{\pm 0.30}$ | $65.53_{\pm 4.09}$ | $65.11_{\pm 0.54}$ |
|  |  | Winkler | $\mathbf{106.07_{\pm 0.47}}$ | $115.73_{\pm 0.20}$ | $105.43_{\pm 0.85}$ | $\mathbf{104.93_{\pm 0.19}}$ |
|  | Transf | $\Delta$Cov | $-0.49_{\pm 0.59}$ | $-1.87_{\pm 0.09}$ | $-1.43_{\pm 1.10}$ | $0.20_{\pm 0.56}$ |
|  |  | PI-Width | $64.06_{\pm 1.74}$ | $59.52_{\pm 0.22}$ | $64.41_{\pm 2.72}$ | $66.49_{\pm 1.17}$ |
|  |  | Winkler | $\mathbf{103.64_{\pm 0.21}}$ | $112.90_{\pm 0.07}$ | $105.97_{\pm 1.21}$ | $\mathbf{105.44_{\pm 0.78}}$ |
|  | ARIMA | $\Delta$Cov | $0.63_{\pm 0.22}$ | $-0.96_{\pm 0.05}$ | $-1.42_{\pm 1.15}$ | $-0.16_{\pm 0.95}$ |
|  |  | PI-Width | $70.43_{\pm 0.86}$ | $67.51_{\pm 0.19}$ | $66.01_{\pm 3.01}$ | $67.33_{\pm 2.28}$ |
|  |  | Winkler | $\mathbf{108.75_{\pm 0.31}}$ | $120.59_{\pm 0.13}$ | $107.20_{\pm 1.21}$ | $\mathbf{103.97_{\pm 0.22}}$ |

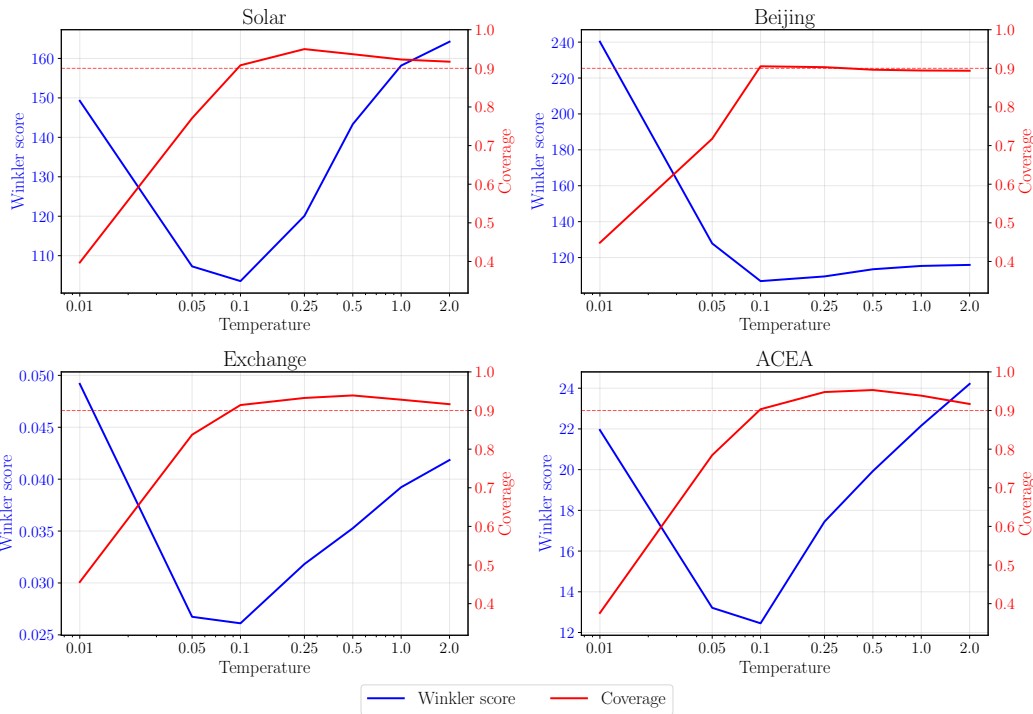

Figure 6: Sensitivity analysis of the temperature.

reservoir space is contained, at the price of a higher Winkler score. For the Beijing dataset, a bigger reservoir dimensionality is required to reach the target coverage.

The sensitivity analysis of the spectral radius can be seen in Fig. 8. It is interesting to see that in the Solar and Beijing datasets, the best performance is achieved even with $\rho(\boldsymbol{W}_h) \geq 1$. This is due to the condition being only sufficient, not necessary: while $\rho(\boldsymbol{W}_h) < 1$ guarantees contractive dynamics in the autonomous linearized reservoir (Gallicchio & Scardapane, 2020), driven reservoirs with input scaling, leaky units, and bounded nonlinearities (Dong et al., 2022; Ceni & Gallicchio, 2025).

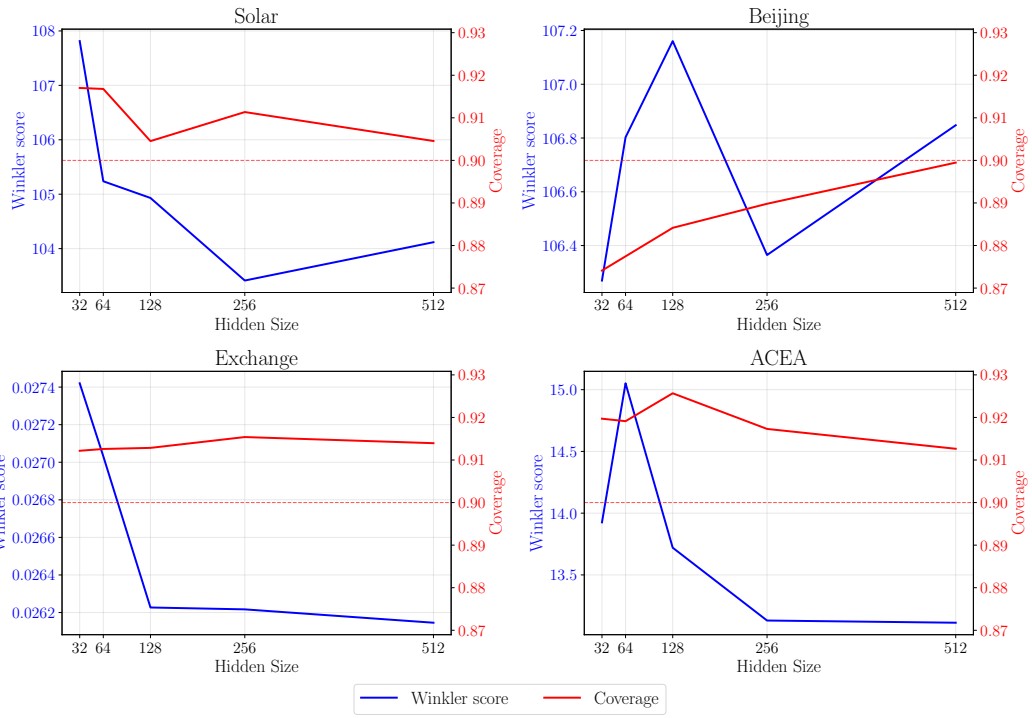

Figure 7: Sensitivity analysis of the reservoir size.

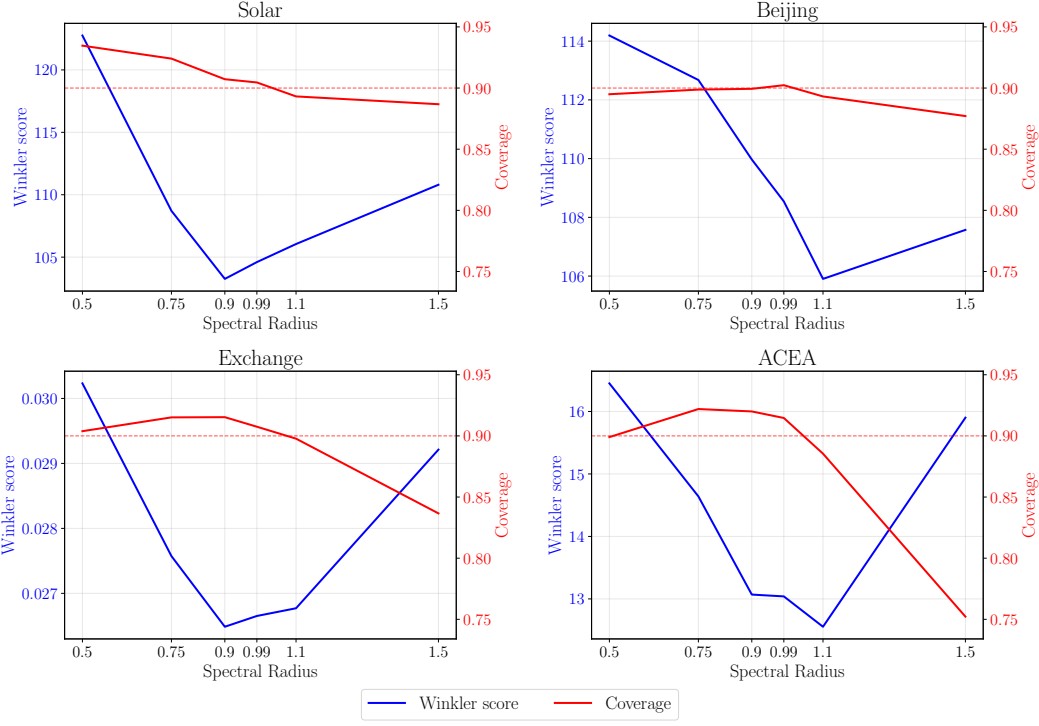

Figure 8: Sensitivity analysis of the spectral radius.

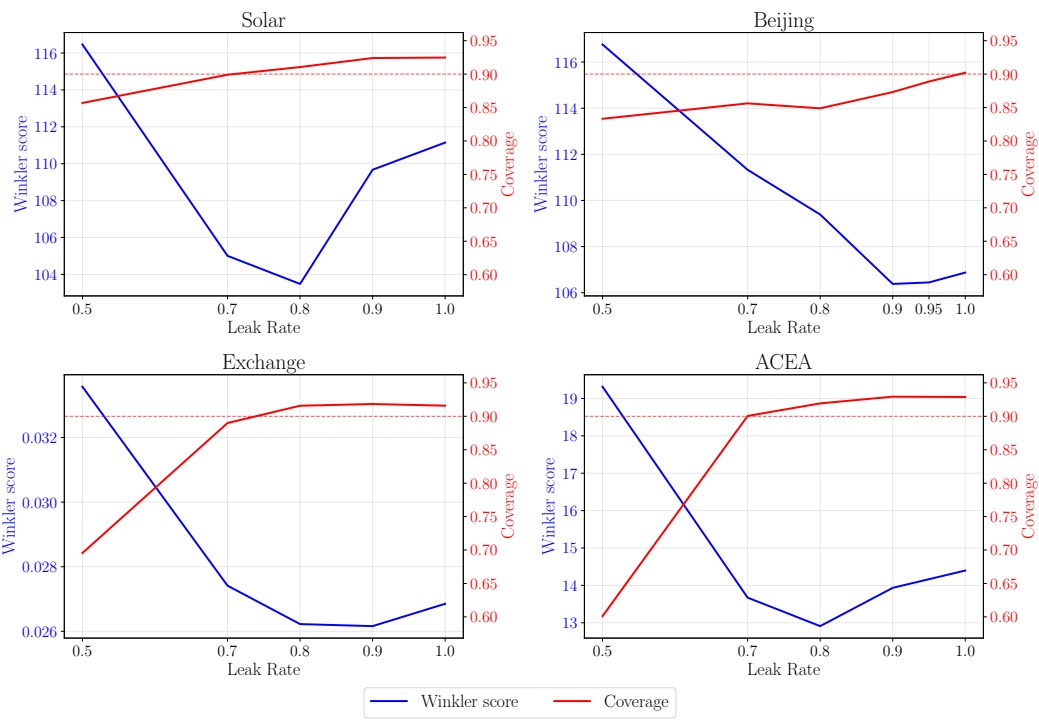

Figure 9: Sensitivity analysis of the leak rate.

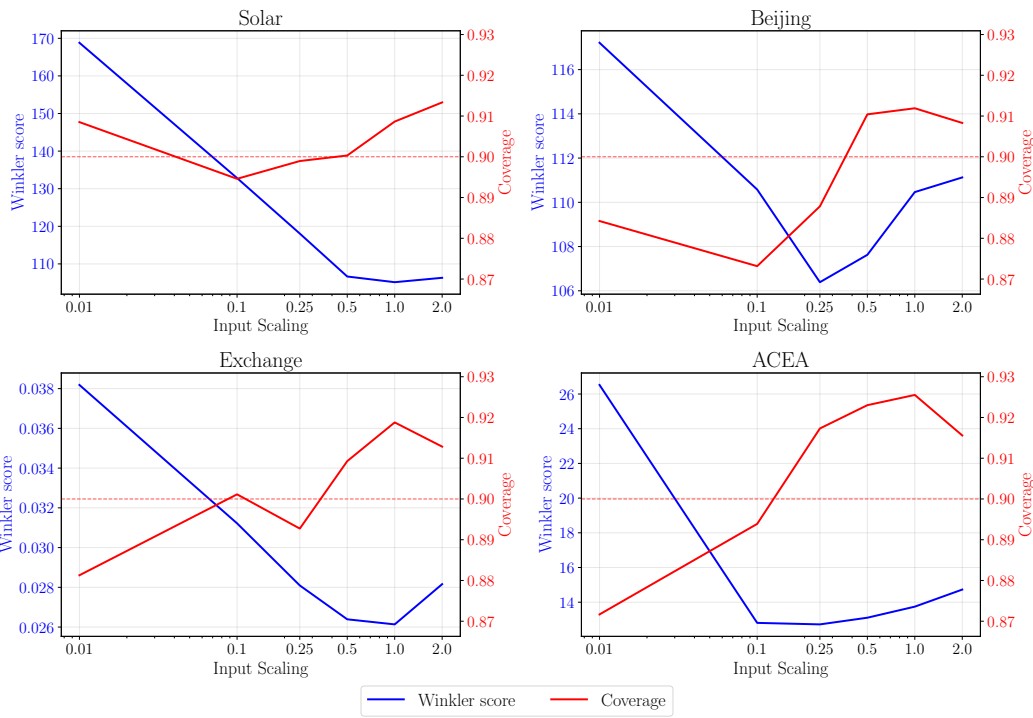

Figure 10: Sensitivity analysis of the input scaling.

In Fig. 9, we observe how varying the leak rate affects the reservoir's performance. In particular, a low value of the leaking rate slows the reservoir's state updates, causing the network to retain past information for longer but reducing its responsiveness to new inputs. We can see how this heavily affects the observed coverage in almost all datasets.

Finally, the effect of different values of input scaling can be found in Fig. 10. High values of input scaling lead the $\tanh$ activation function to saturation during state update, yielding loss of sensitivity to input variations. On the other hand, a small input scaling keeps the activation function in the linear part, limiting the richness of nonlinear dynamics necessary to capture complex temporal patterns.

## G    IMPLEMENTATION DETAILS

The code have been written in Python (Van Rossum & Drake, 2009) using the following open-source packages:

- Numpy (Harris et al., 2020);
- Pandas (The pandas development team, 2020; Wes McKinney, 2010);
- PyTorch (Paszke et al., 2019);
- PyTorch Lightning (Falcon & The PyTorch Lightning team, 2019);
- Torch Spatiotemporal (Cini & Marisca, 2022);
- `reservoir-computing` (Bianchi et al., 2020).

Experiments were conducted on a machine equipped with a AMD Ryzen 9 7900X CPU and a NVIDIA GeForce RTX 4090.

## H    USE OF LARGE LANGUAGE MODELS

We acknowledge the use of Large Language Models as a writing tool for minor edits in single sentences.

## I    ETHICS STATEMENT

The work presented in this paper is about basic machine learning research and we perform experiments on standard, publicly available datasets. The authors have read and adhere to the ICLR Code of Ethics and do not foresee any direct ethical concerns or potential for misuse.

## J    REPRODUCIBILITY STATEMENT

The code to reproduce the experiments presented in this paper will be made publicly available.

