# OpenReview forum: "ResCP: Reservoir Conformal Prediction for Time Series Forecasting"
_ICLR.cc/2026/Conference — ICLR 2026 Poster_

### Official Review · Reviewer_3iSQ · 2025-10-24

**Soundness:** 3
**Presentation:** 4
**Contribution:** 3
**Rating:** 8
**Confidence:** 4

**Summary:**

This paper introduces ResCP (Reservoir Conformal Prediction), a novel training-free method for constructing conformal prediction intervals in time series forecasting. They use Echo State Networks (ESNs) - randomized recurrent neural networks - to compute data-dependent weights for conformal prediction without requiring expensive model training compared to existing works. The authors provide comprehensive theoretical and experimental support for the CP technique.

**Strengths:**

- Excellent presentation. The motivation, algorithm, and results are all very clearly explained and easy to follow.

- The algorithm is strong and fills a needed gap. In 2023 / 2024 there are a bunch of papers that use regression inside the CP algorithm in order to handle patterns in the residual (SPCI and HopCPT are prime examples), although they did achieve better efficiency and are inspiring, the methods are slow and data-hungry, negating some of the advantages of CP itself. Having a high performing training-free (almost non-parametric) alternative is great.

- Strong Empirical Performance. Achieves competitive or superior results across multiple datasets, particularly outperforming HopCPT while being much faster.

**Weaknesses:**

Not much. The only criticism is perhaps the assumptions for ResCP is quite limiting, making the algorithm to be sensitive to data conditions (assumption 3.1) and hyperparameter choices (assumption 3.5). The sensitivity analysis shows performance varies significantly with these choices.

**Questions:**

- Strongly mixing data generation process was not an assumption by SPCI and HopCPT, I believe they both permit distribution shifts or regimes in the data generation process. I also do not think neither weather data (solar and Beijing) nor financial data is strongly mixing? I would love to see a discussion on what kind of data is NOT suitable for ResCP and it performs worse than SPCI for example.

- Condition 3.5 (ii) is quite strong and perhaps need more discussion. How is this clustering property ensured? Is there evidence that the ESN hyperparameters (spectral radius, input scaling, etc.) produce a reservoir that maps similar dynamics to nearby states? Does it place a strong requirement that  the input sequence have strong recurring patterns? The paragraph below assumption 3.5 feels like 3.5(ii) can be a direct result of 3.5(i), which i do not think is the case.

---

> ### Author Response · Authors · 2025-11-20
> **Rebuttal**
>
> Thank you for your review. We are happy that you found our approach interesting and our contribution meaningful! Please find our point-by-point answers to your comments below.
>
> ### Weaknesses
>
> > W1.1) Limiting assumptions
>
> Our theoretical analysis has the purpose of showing that, under reasonable assumptions (time-invariant process and strongly mixing residuals), ResCP is sound. Nonetheless, we also discuss practical approaches to mitigate issues such as non-stationarity (Section 3.1.2), and our experimental evaluation shows that, in practice, ResCP works well in very different settings. See answers to your questions below for more details.
>
> > W1.2) Sensitivity to hyperparameter selection
>
> Sensitivity to hyperparameter selection is a feature of reservoir computing and randomization-based approaches; however, these approaches make up for this limitation with their computational scalability, which makes the cost of running a hyperparameter sweep relatively contained.  Moreover, looking at the sensitivity analysis, it is possible to see that the region of hyperparameter space that leads to good results is quite similar across benchmarks. For example, setting the hidden size 256, the spectral radius to 0.99, the leak rate to 0.8, and input scaling to 0.5 would consistently lead to good results in the considered datasets.
>
> ### Questions
> > Q1) Strongly mixing data generation process was not an assumption by SPCI and HopCPT [...] I also do not think neither weather data (solar and Beijing) nor financial data is strongly mixing? I would love to see a discussion on what kind of data is NOT suitable for ResCP and it performs worse than SPCI for example.
>
> Both HopCPT and SPCI make analogously strong assumptions on the data-generating process. HopCPT allows for the existence of a finite set of error regimes and assumes that the process is stationary within each regime with i.i.d. noise (see Assumptions B.3 and B.4 in [1]). SPCI assumes that the process is stationarity and that dependence among observations decays with time (which is akin to assuming a strongly mixing process – see Assumptions 1 and 2 in [2]). Note that assuming strongly mixing data, i.e., that observations become approximately independent if we wait long enough, although an approximation of reality, is quite standard to enable learning.
>
> The assumptions we made in the theoretical analysis are needed to ensure that the method is theoretically sound when the data-generating process is well-behaved. In practice, and as shown in the experimental evaluation, these requirements are not strictly needed to obtain good performance, and the approach introduced in 3.1.2 can mitigate non-stationarity. In the revision, we added a set of ablation studies (Section 4 and Appendix E) and a synthetic dataset (Appendix D.3) to further investigate these aspects.
>
> > Q2.1) Condition 3.5 (ii) is quite strong and perhaps needs more discussion. How is this clustering property ensured?
>
> Assumption 3.5(ii) can be satisfied by using an appropriate temperature schedule, which makes the weights for states not sufficiently close to the target arbitrarily small. As discussed in the text, this is essentially a bias–variance trade-off.
>
> > Q 2.2) Is there evidence that the ESN hyperparameters (spectral radius, input scaling, etc.) produce a reservoir that maps similar dynamics to nearby states? Does it place a strong requirement that the input sequence have strong recurring patterns?
>
> ESNs indeed map similar dynamics to similar states. If that were not the case, the reservoir would not be able to learn meaningful representations. Acting on the reservoir hyperparameters, it is again possible to trade off bias and variance. For example, a very low spectral radius would map different trajectories to very similar states (very high clustering property) because the resulting reservoir dynamics would be very contractive. Conversely, a high spectral radius would result in more diverse dynamics, making the reservoir more sensitive to differences in its inputs, but increasing the variance (similar time series could be mapped to potentially very different states). For the input scaling, it should be set to avoid saturating the non-linearities (tanh) in the reservoir. We refer to [3] for an in-depth discussion on setting up ESNs effectively.
>
> There are no requirements for recurrent patterns in the input time series; however, without any temporal structure in the residuals, modeling the conditional distribution would not be very meaningful (though this applies to any localized CP method).
>
> ----
> Thank you for the comments. We have updated the discussion to include these considerations.
> ### References
> [1] Auer et al., “Conformal Prediction for Time Series with Modern Hopfield Networks”, NeurIPS 2023
>
> [2] Xu et al., “Sequential Predictive Conformal Inference for Time Series” ICML 2023
>
> [3] Lukoševičius, “A Practical Guide to Applying Echo State Networks”, 2012

---

### Official Review · Reviewer_SLWv · 2025-10-30

**Soundness:** 2
**Presentation:** 2
**Contribution:** 2
**Rating:** 4
**Confidence:** 3

**Summary:**

This study proposed a conformal prediction method for time series forecasting using reservoir computing implemented with echo state network.

**Strengths:**

-	The study utilized reservoir computing implemented by echo state networks, which has not been studied before in the context of conformal prediction for time series.
-	The study provided an asymptotic conditional coverage guarantee and the theoretical analysis was explained thoroughly.
-	Detailed backgrounds and motivation related to the existing studies.

**Weaknesses:**

-	In Assumption 3.1, the author assumed stationarity of $(x_t, r_{t+H})$. I believe the author used residuals for $x_t$ (i.e., $x_t = r_t$) for ResCP, and I think assuming stationarity of residuals (i.e., $(r_t, r_{t+H})$) makes sense. However, in 3.1, the author claimed that “the input $x_t$ can include any set of endogenous and exogenous variables at time step t”.  I am concerned that in this case the assumption does not hold, since in general time series features are not guaranteed to be stationary, which makes conditional coverage somewhat restricted to ResCP with $x_t = r_t$ (since I guess ResCQR utilizes other variables except residuals).


-	The author proposed and used ad-hoc methods, such as using time-dependent weights described in 3.1.2 and handling exogenous inputs (i.e., ResCQR) described in 3.2, on top of the main proposed method (ResCP). However, with the current experiment design, it is difficult to see what is the performance of the proposed method and what is the improvement due to the ad-hoc methods. Based on my understanding, the authors used time-dependent weights as the default to ResCP and used exogenous variables only for ResCQR. Ablation studies regarding these ad-hoc methods will be helpful for readers to understand and interpret the experimental results. For example, ablation study: ResCP vs. ResCP + time-dependent weights vs. ResCP + exogenous variables vs. ResCP + time-dependent weights  + exogenous variables vs. ResCQR.


-	In line 328 under Remark section, the author claimed that “good approximation of F entirely depends on the ability of the ESN to encode all relevant information from the past in its state”. I am not sure what it means since ESN does not have trainable parameters and only has echo state property, which says it asymptotically forgets its initial state. Even if the claim is true, then what is the advantage of using ESN over RNNs or other sequence models? Since I believe RNNs have more capacity to learn “all relevant information from the past in its state” to compute better weights to approximate F (even tho it would not feasible to achieve the asymptotical conditional coverage theoretically as in the way the author derived using other sequence models).


-	The experiment results did not show consistent results supporting the effectiveness of the proposed methods, and the manuscript did not contain enough details about the implementation of the method, experiment setup, to interpret the results (see questions).

**Questions:**

-	In 2.2 Echo State Networks, it is stated that when “$W_h$ and $W_x$ are properly initialized, …”, however throughout the paper, I can only find the condition about $W_h$ which is required to set to satisfy the spectral radius < 1. What is “proper initialization” for $W_x$?

-	In line 217, “we approximate it through Monte Carlo sampling akin to Auer (2023)…”.  I believe its better to describe how the authors did this since its important things to construct the prediction intervals with some details: what sampling size did the authors used? Did authors used sampling size same for all datasets or different size for datasets?

-	In line 225, the authors refined the intervals by selecting $\beta$ that minimizes width. What are $\beta$ candidates did authors used to obtain $\beta^*$ that minimizes interval width?

-	In Assumption 3.2, are there any assumptions need for $W_h$ and $W_x$? how did authors derive the bound for ESN based on $d_{\text{fm}}$

-	What is CP-QRNN baseline? While the authors cited [1], there is no method entitled CP-QRNN in the cited paper. Additionally, the important details of CP-QRNN such as model architecture are missing in the main text and Appendix, which makes difficult to interpret the results since CP-QRNN is the baselines that showed the best performances in the experiments on Beijing and Solar datasets.

-	What exogenous variables are used for the experiments? which methods used exogenous variables?


-	In sec 4.1, the author stated “CP-QRNN baseline achieves strong performance on the large datasets with informative exogenous variables,… but fails in achieving good results in the smaller datasets”. This claim does not sound convincing since we do not know whether exogenous variables are informative. It could be interpreted that in other two datasets where the proposed methods outperformed other baselines,  the exogenous variables are not informative? The authors need to provide details of exogenous variables or carefully designed experiments using synthetic dataset to support this claim. Additionally, what is the definition of the smaller datasets? Based on the dataset details provided in the appendix, the size of the ACEA dataset is the largest (solar data is 26k, Beijing air quality dataset is 35k, exchange dataset is 7.5k, and ACEA dataset is 137k), while the author stated that ACEA dataset is a small dataset.

-	In line 60, the author claimed that “ResCP can be applied on top of any point forecasting model, since it only requires residuals from a disjoint calibration set”. However, isn’t this true for any CP methods? Since technically CP is base prediction model-agnostic.


[1] Relational Conformal Prediction for Correlated Time Series, Cini et al., 2025

---

> ### Author Response · Authors · 2025-11-20
> **Rebuttal (1/3)**
>
> Thank you for your detailed review and precise comments. Please find our point-by-point answer below.
> ###Weaknesses
> > W1) In Assumption 3.1, the author assumed stationarity of $(x_t, r_{t+H})$. [...] However, in 3.1, the author claimed that “the input $x_t$ can include any set of endogenous and exogenous variables at time step t”. I am concerned that in this case the assumption does not hold [...] which makes conditional coverage somewhat restricted to ResCP with $x_t=r_t$.
>
> All theoretical results on the conditional coverage are provided with respect to ResCP (the main approach we propose and focus on in the paper), which, as correctly pointed out by the reviewer, does not use exogenous variables. Moreover, as we do agree with the reviewer that non-stationarity can be a concern in practical applications, Section 3.1.2 discusses practical approaches to mitigate its impact.
>
> We introduce ResCQR as an alternative to ResCP in certain scenarios, as discussed in the paper. The accuracy of the estimate produced by ResCQR will depend on how well the learned regression model can approximate the true quantile function.
> In the updated paper, we provide additional experiments on 1) the impact of using time dependent weights (Section 4 and Appendix E.1) and 2) the impact of adding the exogenous variables in ResCP (Appendix E.2). We also include an additional experiment on synthetic non-stationary data (Appendix D.3) to further assess how these components can deal with non-stationarity. We hope this clarifies your doubts.
>
> > W2) The author proposed and used ad-hoc methods, such as using time-dependent weights described in 3.1.2 and handling exogenous inputs (i.e., ResCQR) described in 3.2, on top of the main proposed method (ResCP). [...] Based on my understanding, the authors used time-dependent weights as the default to ResCP and used exogenous variables only for ResCQR. Ablation studies regarding these ad-hoc methods will be helpful for readers to understand and interpret the experimental results.
>
> Your assumption is correct; we used data-dependent weights with no exogenous variables as the default in ResCP. The appendix already includes a sensitivity analysis. We agree with the reviewer that additional ablation studies are interesting. As already mentioned, we added ablation studies focused on assessing the impact of time-dependent weights and exogenous variables in Section 4 and Appendix E. Results support the adoption of the proposed designs. For your convenience, we also report the results here: https://imgur.com/a/oIHc0P0 and https://imgur.com/9tvU1DY.
>
> > W3) ESN econding capabilities and advantage of using ESN over RNNs or other sequence models.
>
> There might be a misunderstanding; we will try to clarify. Even if ESNs (which are a class of RNNs) do not require any training, it doesn’t mean that they don’t provide a representation of the input dynamics. The representation corresponds to the state of the ESN, and its properties depend on how the ESN is configured, e.g., by setting specific hyperparameters. For example, we can control the sensitivity of the ESN to fast and slow dynamics by tuning its spectral radius or leak rate. Similarly, we can use larger or smaller reservoirs, trading off sample efficiency with the richness of the representation. Indeed, as mentioned in the paper, there is a broad literature discussing the approximation power of ESNs and how to configure them to obtain meaningful representations.
>
> The advantage of a randomized approach over standard RNNs is that the former doesn’t need any training, which results in improved data efficiency and scalability, which is of central importance in the context of conformal prediction as recognized by Reviewer 3iSQ and as we discuss in the paper (see, e.g., Section 1) as a motivation for our approach. Moreover, randomized methods have also been shown to perform well in non-stationary settings (e.g., see [1]). These advantages are empirically demonstrated by comparing with HopCPT, which uses an approach analogous to ResCP but relies on a trainable sequence model (similar to a Transformer).

---

> > ### Author Response · Authors · 2025-11-20
> > **Rebuttal (2/3)**
> >
> > > W4) The experiment results did not show consistent results [..] the manuscript did not contain enough details about the implementation of the method.
> >
> > We respectfully disagree on the lack of supporting results and details. Our results unequivocally show that ResCP consistently achieves performance similar to or better than all baselines, including trained models that require much higher compute or data. Regarding ResCQR, the performance in terms of Winkler is comparable to the RNN-based approach (CP-QRNN), but, compared to the other trainable methods, it is faster and more sample-efficient. Overall, our reservoir-based approach results in CP methods that are accurate and scalable, which are core properties for a CP framework, as discussed in the paper and recognized by the other reviewers.
> >
> > Nonetheless, as already mentioned, we have included additional ablation studies and experiments to further support these claims (Section 4, Appendix D, and Appendix E).  These results further show that ResCP is robust to different settings and compares favorably to the state-of-the-art.
> >
> > ### Questions
> >
> > > Q1) Echo State Networks, it is stated that when “$W_h$ and $W_x$ are properly initialized, …”, however throughout the paper, I can only find the condition about $W_h$ which is required to set to satisfy the spectral radius < 1. What is “proper initialization” for $W_x$?
> >
> > The input weight matrix $W_x$ is initialized with random values drawn from a uniform distribution in the interval [-a, a] where ‘a’ is the input scaling hyperparameter which controls the magnitude of the weights. In particular, input scaling (together with the spectral radius) balances how much the current state depends on the current input and on the previous state. In practice, the input scaling is selected together with the spectral radius based on the task at hand, so that the reservoir produces meaningful dynamics. In particular, the input scaling determines the degree of nonlinearity in the reservoir’s behavior, as it affects the activation of the hidden units. We refer to [2] for an in-depth discussion on how to effectively set up an ESN (which is out of the scope of the paper). We observed that an input scaling of 0.5 consistently resulted in good performance in the considered benchmarks (see sensitivity analysis in the appendix). Thanks for pointing this out.
> >
> > > Q2) Details about MC sampling
> >
> > In all of our experiments, we set the sampling size to the number of observations used for calibration. We provide more details on the sampling procedure in the revised version of the paper (Section 4). Thank you for pointing this out.
> >
> > > Q3) Details about $\beta$ for minimizing PI width
> >
> > In our experiments, we computed quantiles over a linearly spaced set of candidate values ranging from 0 to $\alpha$. Specifically, we used 100 candidate $\beta$ values for all experiments. We also clarify this in the revised version of the paper in Section 4.
> >
> > > Q4) In Assumption 3.2, are there any assumptions need for $W_h$ and $W_x$? how did authors derive the bound for ESN based on $d_{fm}$
> >
> > Assumptions like 3.2 are standard in reservoir computing. Essentially, Assumption 3.2 assumes that the echo-state property holds and that the ESN is contractive (i.e., past inputs are given exponentially lower weight). This can be ensured by setting $||W_h||<1$. While an in-depth discussion is out of the scope of the paper, we refer to several works that discuss these properties at length in the text.
> >
> > > Q5) What is CP-QRNN baseline?
> >
> > CP-QRNN is analogous to the model called CoRNN in [3] and it is simply an RNN trained with a pinball loss to perform quantile regression. The reason we decided to change the name is that “Co” in CoRNN stands for “Correlated” in the context of [3] and wouldn’t make sense in the context of our paper.  We have added a clarification in the paper (Section 4). We have also added details about the CP-QRNN architecture in Appendix C.3.
> >
> > > Q6) What exogenous variables are used for the experiments? which methods used exogenous variables?
> >
> > The methods using the exogenous variables are the trainable ones, i.e., ResCQR, CP-QRNN, HopCPT. The available exogenous variables change from dataset to dataset. In the Solar dataset, besides the solar radiation that we used as the target variable, it also contains 8 other environmental features, such as the air temperature, the wind speed, and the solar zenith angle. In the Beijing dataset, we used the PM10 measurements as the target, and it also contains 10 other environmental variables, including the concentration of atmospheric pollutants.
> >
> > We added in Appendix B the description of each exogenous variable in each dataset. Thank you for pointing this out.

---

> > > ### Author Response · Authors · 2025-11-20
> > > **Rebuttal (3/3)**
> > >
> > > > Q7) Clarification on the size of the datasets
> > >
> > > Thanks for the opportunity to clarify this point better. ACEA is a smaller dataset because, even if the time series has 137k steps, it is just one time series. In Beijing, we have 12 time series, each one of 35k steps (which gives 420k total time steps). Similarly, we have 26*50 = 1300k time steps in the solar datasets. Moreover, since samples in a single time series are not independent, having multiple time series increases the effective sample size more than having a single, longer time series. In addition to that, these datasets have additional exogenous variables that introduce extra information that can be leveraged by models, like CP-QRNN and ResCQR. Regarding the additional details, please refer to the answer to Q6.
> > >
> > > > Q8) In line 60, the author claimed that “ResCP can be applied on top of any point forecasting model, since it only requires residuals from a disjoint calibration set”. However, isn’t this true for any CP methods?
> > >
> > > This is correct for most CP methods. In line 60, we wanted to highlight one of the main advantages of adapting a CP framework for uncertainty quantification. However, note that, differently from ResCP, several CP that try to localize uncertainty estimates rely on the base predictor already providing some form of uncertainty estimates (see, e.g., [4]).
> > >
> > > ----
> > > We hope that our answers and the additional results can clarify your doubts. If there’s anything else preventing you from recommending a higher score, please let us know. We would be happy to discuss more.
> > >
> > > ### References
> > > [1]Prabhu et al., “RanDumb: Random Representations Outperform Online Continually Learned Representations”, NeurIPS 2024
> > >
> > > [2] Lukoševičius, “A Practical Guide to Applying Echo State Networks”, 2012
> > >
> > > [3] Cini et al., “Relational Conformal Prediction for Correlated Time Series”, ICML 2025
> > >
> > > [4] Romano et al., “Conformalized quantile regression”, NeurIPS 2019

---

### Official Review · Reviewer_TvuY · 2025-10-31

**Soundness:** 3
**Presentation:** 3
**Contribution:** 2
**Rating:** 6
**Confidence:** 3

**Summary:**

The paper studies prediction set construction for time-series data.
The proposed method builds on reservoir computing and learns the
residual distribution by dynamically weighting the calibration samples.
The method is evaluated on real datasets, showing advantages in
coverage and computation efficiency.

**Strengths:**

1. The paper is well-written and easy to follow.
2. Integrating conformal prediction with reservoir computing demonstrates the potential to unite statistical efficiency and computational efficiency in predictive analysis of time-series data—achieving the best of both worlds.

**Weaknesses:**

1. The theoretical results are relatively clean, but depend on a set of assumptions.
It would be helpful to evaluate the method in synthetic environments (where the data-generating process is fully known)
and systematically investigate the method's sensitivity to violations of its assumptions.

2. In the numerical results, it appears that CP-QRNN often achieves strong statistical performance in
reasonable computing time, compared with ResCQR. I wonder if the authors could provide more discussion on
the comparison, to clarify better the advantage of ResCQR and how a practitioner should choose between the methods
in different scenarios.

**Questions:**

Please refer to the "Weaknesses" section.

---

> ### Author Response · Authors · 2025-11-20
> **Rebuttal**
>
> Thank you for reviewing our work and for the useful suggestions. We really appreciate that.
>
> ### Weaknesses
>
> > W1) Evaluation in synthetic environments and violations of the assumptions
>
> Thank you for your suggestion. In the revised manuscript, we added a new experiment to the appendix where we evaluate ResCP on a synthetic dataset with non-stationary time series generated by an AR process whose coefficients change over time. The results show that ResCP attains the desired coverage in this setting, with the designs introduced in 3.1.2 being an essential component to deal with non-stationarity..The table can be found in Appendix D.3 and, for your convenience, we also reproduce it here: https://imgur.com/a/XeNfiI0.
>
> Regarding possible violations of the temperature scheduling assumption (Assumption 3.5), we refer you to the sensitivity analysis in Appendix D, Figure 3, in which it can be seen that if the temperature is too small, the method fails to achieve the nominal coverage (red curve) on all datasets. If, on the other hand, the temperature is too high, the method can achieve the desired coverage, but the Winkler metric gets worse due to larger intervals.
>
> Finally, we have added a set of ablation studies (Section 4/Appendix E) to further analyze the impact of each design in obtaining good performance in different settings.
>
> > W2) Advantages of ResCQR vs. CP-QRNN
>
> ResCQR takes approximately half of the training time required by CP-QRNN, even less in several scenarios. Indeed, in ResCQR, the linear readout is the only trainable part of the model, which makes the model extremely scalable (can be easily trained on a CPU). This also means that the model can be more sample-efficient due to the lower number of trainable parameters.
>
> Finally, we would like to stress that the ResCQR approach is not the focus of our work; it is a variant of ResCP that we introduce to show how exogenous inputs can be handled within a reservoir computing approach for CP. Future work could further expand upon ResCQR-like approaches, e.g., by making the linear readout adapt to changes over time.
>
>
> Thank you again for your comments. Please let us know if there are any doubts requiring further clarification.

---

### Official Review · Reviewer_WWw1 · 2025-11-01

**Soundness:** 2
**Presentation:** 3
**Contribution:** 2
**Rating:** 2
**Confidence:** 4

**Summary:**

The work proposes a novel method that utilises Reservoir Computing to reweight conformity scores. This helps account for the local temporal dynamics, and under some assumptions, the method can achieve conditional coverage. Empirical evidence for the method's efficacy is provided across various forecasting tasks.

**Strengths:**

The idea is quite interesting to use Reservoir Computing to adapt the residuals. The background is well-written and thorough. The experiments are also conducted with various datasets, baselines, and models. The experiments are repeated across several runs, giving a good picture of the method's stability.

**Weaknesses:**

The biggest weakness is the comparison with only one significance level. To best judge a conformal method, it is imperative to compare its performance across different significance levels, and plotting a calibration curve is particularly helpful.

The notations are unclear at times and the presentation could be improved.

See "Questions*" below for more

**Questions:**

1. Line 078: If I understand correctly, y_t is a scalar observation. How would the method incorporate the cases where we have multidimensional time series forecasting?

2. Line 081: If I understand correctly, y_hat_{t+H} is a vector corresponing to H steps of prediciton. Since both times, a small y is used, it becomes notationally confusing to follow the stuff here.

3. In equation 2, shouldn't it be the absolute residuals, since they are used as the nonconformity scores for regression?

4. Line 102: "To address heteroskedasticity": does it refer to heteroskedasticity over the values, or time index, or both?

5. Line 191: Is x_t supposed to be the residual, not the history i.e y_{t-1}...

6. Line 196: Is x the time series now instead of y, and not the exogenous variables or residuals?

7. Equation 6: Shouldn't s be upper-bounded by T and not T-H?

8. Equation 11: If we want symmetric intervals, can we work with absolute residuals and adapt them accordingly?

9. Assumption 3.1: Earlier, it was alluded to that under mild assumptions, the coverage is satisfied; however, strongly mixing seems to be a rather strong condition. Would any of the real-world time series even satisfy it?

10. "Assumption 3.2 tells us that the reservoir state evolves in a stable and predictable manner"  This also seems a strong assumption, especially if there is a change of distribution.

11. Line 338 "do not include any positional encoding": Could having a positional encoding help the algorithm?

12. Regarding Table 1: Firstly, it is interesting that adding exogenous variables is hurting the performance, as apparent from RESCQR. As for RESCP, it is not clear if there are huge advantages; in most cases, across learning and non-learning, the average width is better for Learning methods, whereas many of the learning methods still have better Winkler score.

13. The target confidence is set as high as 0.9, which is okay. However, it is necessary to demonstrate the performance of the proposed method with varying significance levels. A calibration curve may be helpful in determining if the proposed method works effectively in all cases.

14. The theoretical statements are there for conditional coverage, but there is no empirical evidence for the same.

---

> ### Author Response · Authors · 2025-11-20
> **Rebuttal (1/3)**
>
> Dear Reviewer WWw1,
> Thank you for your review and suggestions. We address all your concerns and questions in the following and remain available for further clarifications.
> ### Weaknesses
> > W1) The biggest weakness is the comparison with only one significance level. To best judge a conformal method [...] plotting a calibration curve is particularly helpful.
>
> We performed other experiments for both our methods and baselines with different significance levels across multiple runs and added them to the paper. The results are consistent and the method is reliable. We included the new results in the revised paper in Section 4 and Appendix E. For your convenience, you can also see the results at the following links:
> - Comparison with the proposed methods and baselines across all datasets and using the an RNN as base model: https://imgur.com/a/cLWKUpb
> - More plots over all dataset/base model combinations for the proposed methods: https://imgur.com/a/uWGv1Cz
> - Additional results at different miscoverage levels: https://imgur.com/a/bL8Lq6z and https://imgur.com/a/ULZohiL.
>
> > W2) The notations are unclear at times and the presentation could be improved. See "Questions*" below for more.
>
> Thanks for pointing that out, we improved and polished the paper as per your suggestions (see below).
> ### Questions
> > Q1) Line 078: If I understand correctly, $y_t$ is a scalar observation. How would the method incorporate the cases where we have multidimensional time series forecasting?
>
> Yes, in our work $y_t$ is a scalar. While modeling joint probabilities within our framework is certainly interesting, it is a different (non-trivial) orthogonal problem that is out of scope for this work. As mentioned in the conclusion, future work could, in particular, explore how to combine ResCP with methods for multidimensional CP discussed, e.g., in [1,2].
>
> > Q2) Line 081: If I understand correctly, $\hat{y}_{t+H}$ is a vector corresponing to H steps of prediciton. Since both times, a small $y$ is used, it becomes notationally confusing to follow the stuff here.
>
> We believe there is a misunderstanding here. $\hat{y}_{t+H}$ is a scalar value, corresponding to the point forecast at time step $t+H$.
>
> > Q3) In equation 2, shouldn't it be the absolute residuals, since they are used as the nonconformity scores for regression?
>
> We used raw residuals so that the PIs that the method builds can be asymmetric.
>
> > Q4) Line 102: "To address heteroskedasticity": does it refer to heteroskedasticity over the values, or time index, or both?
>
> In that specific sentence, we mean heteroskedasticity in terms of variability of the residuals conditional on the (recent) past observations, hence over values.
>
> > Q5) Line 191: Is $x_t$ supposed to be the residual, not the history i.e $y_{t-1}$...
>
> Apologies, but we do not understand the issue here. In line 191 we indeed write that $x_t$ is the residual.
>
> > Q6) Line 196: Is $x$ the time series now instead of $y$, and not the exogenous variables or residuals?
>
> We believe there is a misunderstanding. Throughout the manuscript, $y$ is always the target, whereas, as we state in lines 190-191, $x$ is the input upon which we condition the uncertainty estimates. In practice, this corresponds to residuals for ResCP, while it might also include exogenous variables and the time series itself for ResCQR.
>
> > Q7) Equation 6: Shouldn't $s$ be upper-bounded by $T$ and not $T-H$?
>
> No, the text in the paper is correct. We upper bound $s$ by $T-H$ because any calibration state $h_s$​ must be matched with a known $H$-step-ahead residual. If you have observations up to time $T$, the last state associated with an available residual is at $T-H$. Hence, the calibration pool is $s=1,\dots,T-H$.
>
> > Q8) Equation 11: If we want symmetric intervals, can we work with absolute residuals and adapt them accordingly?
>
> Yes, absolutely. Instead of taking two quantiles (i.e., $\hat{q}^{\alpha/2}, \hat{q}^{1-\alpha/2}$), it would be enough to take a single quantile ($\hat{q}^{1-\alpha}$) and build the intervals accordingly ($[\hat{y} - \hat{q}^{1-\alpha}, \hat{y} + \hat{q}^{1-\alpha} ]$).

---

> ### Author Response · Authors · 2025-11-20
> **Rebuttal (2/3)**
>
> > Q9) Assumption 3.1: Earlier, it was alluded to that under mild assumptions, the coverage is satisfied; however, strongly mixing seems to be a rather strong condition. Would any of the real-world time series even satisfy it?
>
> Assuming stationarity and a strongly mixing process is quite standard in CP methods for time series and learning methods for sequential data in general. For example,  both HopCPT and SPCI make similar assumptions. Note that assuming strongly mixing data, i.e., that observations become approximately independent if we wait long enough, although an approximation of reality, is quite standard to enable learning. As already mentioned, assuming that a process is strongly mixing is a very common and well-established approach in statistics to derive asymptotic properties when working with time series data. Finally, note that we never state that any of the assumptions are mild; we say that they are reasonable in the context of our work.
>
> > Q10) "Assumption 3.2 tells us that the reservoir state evolves in a stable and predictable manner" This also seems a strong assumption, especially if there is a change of distribution.
>
> Assumptions like 3.2 are standard in reservoir computing. Essentially, Assumption 3.2 assumes that the echo-state property holds and that the ESN is contractive (i.e., past inputs are given exponentially lower weight). This is a property of the state: it can be ensured by setting $||W_h||<1$, and doesn’t imply strong assumptions on the input dynamics. We refer to several works that discuss these properties at length in the text.
>
> We agree that “predictable” might create some misunderstanding and we will clarify better in the revision.
>
> > Q11) Line 338 "do not include any positional encoding": Could having a positional encoding help the algorithm?
>
> In line 338, we use the term “positional encoding” to indicate a generic mechanism to account for the temporal distance between residuals at two different time steps. We discuss how to implement such a mechanism in 3.1.2. Simply concatenating an arbitrary positional encoding to the input wouldn’t necessarily result in the desired behavior and would be redundant given the time-dependent weights introduced in 3.1.2.
>
> > Q12) Regarding Table 1: Firstly, it is interesting that adding exogenous variables is hurting the performance, as apparent from RESCQR. As for RESCP, it is not clear if there are huge advantages; in most cases, across learning and non-learning, the average width is better for learning methods, whereas many of the learning methods still have better Winkler score.
>
> There might be a misunderstanding here; we are not quite sure about the point being raised. ResCP performs on par with or better than all the considered baselines (learning or non-learning) in all settings, except possibly for Solar, where covariates indeed play a significant role. Moreover, ResCP is much more scalable and robust as shown by its computational cost and the performance on the smaller datasets. We believe the results unequivocally show these advantages. Note that the metrics reported in the table cannot be evaluated in isolation. For instance, one can achieve good coverage simply by using arbitrarily wide intervals; conversely, another method may produce very narrow intervals that fail to provide the required coverage. The challenge is to achieve good coverage while maintaining a reasonably small interval width.
>
> As already mentioned, to further improve our analysis, we have added an ablation study (Section 4, Appendix E), and an analysis of performance on a synthetic non-stationary dataset (Appendix D.3).
>
> > Q13) The target confidence is set as high as 0.9, which is okay. However, it is necessary to demonstrate the performance of the proposed method with varying significance levels. A calibration curve may be helpful in determining if the proposed method works effectively in all cases.
>
> As already mentioned, we updated the paper to include calibration curves over multiple significance levels in Section 4 and Appendix D. In the main text, we added a plot that shows a comparison of the proposed methods against a selection of baselines (you can see it here https://imgur.com/a/cLWKUpb). We also added a more complete analysis in Appendix E (which you can also see here https://imgur.com/a/uWGv1Cz). Note that while NexCP appears to be well calibrated, it does not provide any mechanism to adapt uncertainty estimates to the local characteristics of the data, and it consequently produces much wider intervals (often twice the length of those produced by ResCP).

---

> ### Author Response · Authors · 2025-11-20
> **Rebuttal (3/3)**
>
> > Q14) The theoretical statements are there for conditional coverage, but there is no empirical evidence for the same.
>
> Empirically quantifying conditional coverage is not trivial, as you get only one sample for any given input. One indication that the approach can adapt to the local characteristics of the data is the fact that ResCP achieves the required coverage while managing to provide prediction intervals with a much smaller average width compared to, e.g., non-adaptive methods such as SCP and NexCP.
>
> To provide an additional assessment of this local adaptability, we have added a qualitative analysis in Appendix D.4 where we show how the width of the intervals changes with the underlying time series dynamics. We also report the plots here for your convenience: https://imgur.com/a/IQzfWE6 and https://imgur.com/a/OgyKxgW.
>
> ### References
> [1] Dheur et al., “A Unified Comparative Study with Generalized Conformity Scores for Multi-Output Conformal Regression” ICML 2025
>
> [2] Feldman el al., “Calibrated Multiple-Output Quantile Regression with Representation Learning” JMLR 2023

---

> > ### Comment · Reviewer_WWw1 · 2025-11-28
> >
> > Thanks for writing the rebuttal. I have some follow up questions.
> >
> > 1.  "While modelling joint probabilities within our framework is certainly interesting, it is a different (non-trivial) orthogonal problem that is out of scope for this work." - Could the authors elaborate more in detail why they see it as a non-trivial problem? Traditionally, multivariate CP can be done by simply finding a mapping of conformity scores to a scalar. What challenges do they see in easily adapting their method to multivariate cases?
> >
> > 2. "We believe there is a misunderstanding here. $\hat{y}_{t+H}$ is a scalar value, corresponding to the point forecast at time step $t+H$" - Just to clarify there is no multi-step prediction here, but a prediction for a single step some time in future, right?
> >
> > 3. "In line 191 we indeed write that $x_t$ is the residual": this is true, but I believe "r" was also used as residual earlier that made it a bit confusing.
> >
> > 4. Could the authors write down the mathematical definition of strongly-mixing? Does simple processes with a linear trend or regular seasonality exhibit strongly-mixing property?
> >
> > 5. "We discuss how to implement such a mechanism in 3.1.2." Please correct me if I missed it in the main paper. The ablation or the effect of positional encoding is not discussed, is it?
> >
> > 6. Varying significance levels - thanks for running the experiments for different significance levels. I noticed RES-CQR has some undercoverage in several experiments,  whereas RES-CP works quite well. Could the authors comment on this behaviour?
> >
> > 7. I am happy to see the calibration plot, but I wish there were more levels there. Some conformal methods show better coverages when low significance levels is chosen but not when it is higher. Stability of the method can be best gauged with some higher levels. Nonetheless, I would take the word of the authors on the stability if they run any experiments on much higher significance levels than the ones reported in the paper.
> >
> > 8. Conditional coverage: Indeed it is hard to compute it, but it could be easily demonstrated on the simulated data. However, thanks for the conditional coverage experiments that you ran for the rebuttal.

---

> > > ### Author Response · Authors · 2025-12-01
> > > **Response to follow-up questions**
> > >
> > > Thank you for the additional feedback. We are happy to have addressed your concerns. Please find point-by-point answers to your additional questions below.
> > >
> > > > Q1) Multivariate CP can be done by simply finding a mapping of conformity scores to a scalar. What challenges do they see in easily adapting their method to multivariate cases?
> > >
> > > Indeed, multidimensional CP methods often rely on mappings to a scalar. Still, such mappings can be far from trivial if one wishes to obtain flexible confidence sets (e.g., recent approaches rely on invertible latent variable models [1] and/or directional quantile regression [2]). What we mean is: this is a different orthogonal problem with a lot of active current research. Adequately covering also this aspect in a single conference paper would not be possible. One could use ResCP to reweight scores also in the case of multidimensional CP, and this would likely work reasonably well; nonetheless, this is out of scope here. Another possibility is simply to estimate a separate PI for each dimension. We will further discuss this aspect as possible future work in the conclusion of the paper.
> > >
> > > > Q2) Just to clarify there is no multi-step prediction here, but a prediction for a single step some time in future, right?
> > >
> > > Yes, we used just a single step.
> > >
> > > > Q3) I believe "r" was also used as residual earlier that made it a bit confusing.
> > >
> > > With $x$ we indicate the input of the reservoir. The residual $r$ can be such an input. We will further clarify this in the text.
> > >
> > > > Q4) Could the authors write down the mathematical definition of strongly-mixing? Does simple processes with a linear trend or regular seasonality exhibit strongly-mixing property?
> > >
> > > A stochastic process $(X_t)_{t \in \mathbb{Z}}$ is called strongly mixing (or $\alpha$-mixing) if its strong mixing coefficients
> > >
> > > $
> > > \alpha(n) = \sup_{k \in \mathbb{Z}}
> > > \sup_{A \in \sigma(X_t : t \le k), B \in \sigma(X_t : t \ge k+n)}
> > > \big|\mathbb{P}(A \cap B) - \mathbb{P}(A)\mathbb{P}(B)\big|
> > > $
> > >
> > > satisfy
> > >
> > > $
> > > \alpha(n) \xrightarrow[n \to \infty]{} 0.
> > > $
> > >
> > > Intuitively, a process is strongly mixing if observations are asymptotically independent.
> > >
> > > If a process is strongly mixing, adding a simple deterministic trend/stationarity on top of it does not change things (as the random component of the process is unchanged).
> > >
> > > > Q5 The ablation or the effect of positional encoding is not discussed, is it?
> > >
> > > We performed an ablation study of the mechanism introduced in 3.1.2 in Section 4.1 (Table 3) and Appendix E.1 (Table 8) and used sinusoidal datetime encodings as additional exogenous variables in the ablation study in Appendix E.2 (Table 9). However, we stress again that this is very different from the positional encodings used, e.g., in Transformers, since, as we explained, it would not make much sense to include them in the architecture.
> > >
> > > > Q6) I noticed ResCQR has some undercoverage in several experiments, whereas ResCP works quite well. Could the authors comment on this behaviour?
> > >
> > > As we discuss in Section 4.1, this might be due to several reasons; however, we observed that most trainable methods obtain worse performance in those datasets. This is likely due to both sample-efficiency issues and the presence of shifts that would require the trained models to be updated over time.
> > >
> > > > Q7) I wish there were more levels there. Some conformal methods show better coverages when low significance levels is chosen but not when it is higher.
> > >
> > > We have a range of confidence intervals that are most useful and often used in practical applications, as using a higher $\alpha$ would result in <75% confidence. However, we have run additional experiments for additional levels. See the results here: https://imgur.com/a/HlhO5uN. As we can see, results are consistent across significance levels.
> > >
> > > > Q8) Conditional coverage: Indeed it is hard to compute it, but it could be easily demonstrated on the simulated data.
> > >
> > > The experiment on synthetic data (Appendix D.3, Table 7) indeed does show that ResCP can adapt to local changes in the data-generating process. We believe the qualitative results in Appendix D.4 (Figures 4 and 5) are the most significant in showing local adaptivity in practical settings.
> > >
> > > -----
> > > References:
> > >
> > > [1] Dheur et al., “A Unified Comparative Study with Generalized Conformity Scores for Multi-Output Conformal Regression” ICML 2025
> > >
> > > [2] Feldman el al., “Calibrated Multiple-Output Quantile Regression with Representation Learning” JMLR 2023

---

### Author Response · Authors · 2025-11-20
**General comments and change list**

We thank all the reviewers for the valuable feedback. We addressed all the questions and concerns in the following. In addition, we made several modifications and additions to the paper, which we summarize below:
- Following the suggestions of reviewer WWw1, we added the calibration curves of ResCP, ResCQR, CP-QRNN and NexCP over all datasets with the RNN baseline (Fig. 2, main body) and the calibration curves of ResCP and ResCQR over all datasets and baselines (Fig. 8). Moreover, Appendix D now includes tables with full results across additional coverage levels.
- As suggested by reviewer SLWv, we added two new ablation studies assessing 1) the effectiveness of the design introduced in 3.1.2 to deal with non-stationarity (Section 4 and Appendix E), and 2) the use of exogenous variables (Appendix E).
- As recommended by reviewer TvuY we added an experiment on a synthetic non-stationary time series generated by multiple AR processes with varying parameters (Appendix D).
- By taking reviewers’ feedback into account, we updated the text to include clarifications when needed (updated text is in green).

---

### Author Response · Authors · 2025-11-26
**Looking forward to your feedback**

Dear Reviewers,

Thank you again for reviewing our paper! Our detailed rebuttals and the revised manuscript address all the concerns and comments raised. We were wondering if you have any concerns left and are looking forward to your further feedback.

Best regards,

The Authors

---

### Author Response · Authors · 2025-12-03
**Summary of Rebuttal Changes and Discussions**

Dear Area Chair,

We attach a high-level summary of our rebuttal and discussion with the reviewers. From the start, **all** reviewers appreciated the **novelty** and **relevance** of our contribution. All concerns raised in the initial reviews were about clarifications, additional ablations, and sensitivity analysis. We have addressed all requests in the revised version of the paper (see the change list in the general comment).

- **Reviewer 3iSQ (initial score: 8)**. The reviewer had mainly questions on the assumptions in our theoretical analysis and sensitivity to hyperparameters. We clarified that ResCP’s assumptions are reasonable and no stricter than those of existing methods, and discussed sensitivity to hyperparameters, referencing the sensitivity analysis in the appendix. To support the discussion, we added additional ablations in **Appendix E** and experiments in a controlled environment in **Appendix D.3**.
- **Reviewer SLWv (initial score: 4)**. The main request was to provide additional ablations. To address this, we added several comprehensive ablation studies (**Section 4** and **Appendix E.1–E.2**) on each component of our approach. We also included results on a synthetic dataset (**Appendix D.3**), to show that the proposed designs can help in non-stationary settings. Moreover, following the reviewer’s suggestion, we added more details on the implementation and datasets in the appendices (**Appendix B.1, C.3, G**). Finally, we clarified the reviewer’s doubts: we discussed the theoretical guarantees that apply specifically to ResCP and how to initialize the reservoir to obtain the desired behavior and ensure that assumptions are met. We also provided more discussion on the empirical results and the flexibility of the proposed method.
- **Reviewer TvuY (initial score 6)**. The reviewer had only two questions related to (1) sensitivity to the violations of theoretical assumptions and (2) the advantages of a variant of the proposed approach (ResCQR).  For (1), we clarified how to ensure that assumptions are met and how hyperparameters control properties of the model, providing a practical recipe. As suggested by the reviewer, we added experiments on synthetic data (**Appendix D.3**) to show the ResCP’s performance when stationarity assumptions are violated. Moreover, we added ablation studies in **Section 4** and **Appendix E** to show how specific design choices affect performance across datasets. For (2),  we clarified that the advantages of ResCQR are significant in terms of training time (requiring roughly half the time) and sample efficiency (simpler model with fewer parameters). Finally, we clarified that ResCQR is a variant for incorporating exogenous inputs, rather than the primary contribution.
- **Reviewer WWw1 (initial score 2)**: The main concern raised by the reviewer was that the uncertainty quantification results were reported only for a single confidence level. To address this, we added several experiments. The new results, which confirm the effectiveness of our method, are shown in **calibration curves** comparing ResCP to baselines (*Section 4 – Figure 2** and **Appendix D.2 – Figure 3**), as well as in tables reporting complete metrics in **Appendix D.1 (Tables 5 and 6)**. In the follow-up questions, the reviewer asked us to check performance at lower confidence levels; we ran additional experiments and confirmed the robustness of our method (see our response to the reviewer’s follow-up). To address the reviewer’s additional questions, we incorporated in the paper several clarifications regarding the notation, the validity of our assumptions, and how these assumptions can be satisfied in practice. Finally, we added further results showing that our approach can effectively localize predictions and provide approximate conditional coverage (see the analysis in **Appendix D.4**, the ablations in **Appendix E**, and the experiment on synthetic data in **Appendix D.3**). We believe that their concerns are now fully addressed.


We hope that this summary can be useful, and we really thank you for handling our paper in these difficult circumstances.

---

### Meta-Review · Area_Chair_m9QM · 2026-01-10

**Summary:**

The reviewers acknowledge the novelty of the paper but raised several questions about: i) the practicality of the assumptions; ii) the sensitivity of the approach to different natures of the datasets and violation of the assumptions. In addition, there were several clarification questions which, as shown in the rebuttal, where reasonably easy to answer.

In response to the reviewers' concerns the authors have elaborated on thesis theoretical assumptions and run several ablation studies to show the robustness of the proposed method. They have also extended the Appendix to include further details on the experiments and new results, including the requested calibration curves.

**Reviewer Concerns:**

I believe that the reviewers' concerns have been properly addressed and the authors have significantly improved the paper during the rebuttal period. The resulting paper is solid and, in my opinion above the acceptance bar.

**Reviewer Scores:**

While the reviewers have not explicitly stated their intention about the scores, I believe the engagement of the most negative assessments show that the discussion was indeed going in the right direction. I have read all the answers in detail and believe that no major concern remain. Also, the depth of the discussion shows that the paper started already in a good state and now, after the rebuttal, contain all the bits necessary for an ICLR publication.

---

### Decision · Program_Chairs · 2026-01-26

Accept (Poster)